# DNA double-strand break repair pathway regulates PD-L1 expression in cancer cells

Hiro Sato[1], Atsuko Niimi[2], Takaaki Yasuhara [3], Tiara Bunga Mayang Permata[1], Yoshihiko Hagiwara[1], Mayu Isono[4], Endang Nuryadi[1], Ryota Sekine[4], Takahiro Oike[1], Sangeeta Kakoti[1], Yuya Yoshimoto[1], Kathryn D. Held[5,6], Yoshiyuki Suzuki[7], Koji Kono[8], Kiyoshi Miyagawa[3], Takashi Nakano[1,2] & Atsushi Shibata[4,9]

Accumulating evidence suggests that exogenous cellular stress induces PD-L1 upregulation in cancer. A DNA double-strand break (DSB) is the most critical type of genotoxic stress, but the involvement of DSB repair in PD-L1 expression has not been investigated. Here we show that PD-L1 expression in cancer cells is upregulated in response to DSBs. This upregulation requires ATM/ATR/Chk1 kinases. Using an siRNA library targeting DSB repair genes, we discover that BRCA2 depletion enhances Chk1-dependent PD-L1 upregulation after X-rays or PARP inhibition. In addition, we show that Ku70/80 depletion substantially enhances PD-L1 upregulation after X-rays. The upregulation by Ku80 depletion requires Chk1 activation following DNA end-resection by Exonuclease 1. DSBs activate STAT1 and STAT3 signalling, and IRF1 is required for DSB-dependent PD-L1 upregulation. Thus, our findings reveal the involvement of DSB repair in PD-L1 expression and provide mechanistic insight into how PD-L1 expression is regulated after DSBs.

[1] Department of Radiation Oncology, Gunma University Graduate School of Medicine, Maebashi, Gunma 371-8511, Japan. [2] Research Program for Heavy Ion Therapy, Division of Integrated Oncology Research,  Gunma University Initiative for Advanced Research (GIAR), Maebashi, Gunma 371-8511, Japan. [3] Laboratory of Molecular Radiology, Center for Disease Biology and Integrative Medicine, Graduate School of Medicine,  The University of Tokyo, Bunkyo-ku, Tokyo 113-8655, Japan. [4] Advanced Scientific Research Leaders Development Unit, Gunma University, Maebashi, Gunma 371-8511, Japan. [5] Department of Radiation Oncology, Massachusetts General Hospital/Harvard Medical School, Boston, MA 02114, USA. [6] International Open Laboratory, Gunma University Initiative for Advanced Research (GIAR), Maebashi, Gunma 371-8511, Japan. [7] Department of Radiation Oncology, Fukushima Medical University, Fukushima 960-1295, Japan. [8] Department of Gastrointestinal Tract Surgery, Fukushima Medical University, Fukushima 960-1295, Japan. [9] Present address: Education and Research Support Center, Graduate School of Medicine, Gunma University, Maebashi, Gunma 371-8511, Japan. Correspondence and requests for materials should be addressed to A.S. (email: shibata.at@gunma-u.ac.jp)

Programmed cell death-1 (PD-1) is an immune receptor, which is expressed on activated CD4+ T cells and CD8+ T cells as well as on B cells in the periphery[1]. PD-1 has a role to inhibit T-cell proliferation and interferon-gamma (IFNγ) production in T cells[2]. Programmed death-ligand 1 (PD-L1) is identified as the ligand of PD-1[3]. The major role of PD-1/PD-L1 interaction is to regulate autoimmune response in the peripheral tissue; PD-1-deficient mice show hyperactivation of the immune system[2]. Further, mounting evidence showed that PD-1 or PD-L1 deficient mice developed different autoimmune diseases in each genetic background[2]. Alternatively, the downregulation of PD-1/PD-L1 interaction can be a cause of human autoimmune disease. The association was reported between single-nucleotide polymorphisms (SNPs) on human PD-1 gene or augmented PD-1 expression and human autoimmune diseases[4]. Thus, the interaction between PD-1 and PD-L1 is critical to control a balanced global immune response in the human body. In addition, PD-1 has roles in cellular response; for example, PD-1 is involved in the immune reaction against HIV infection[5].

Since decades ago, several distinct strategies for cancer immune therapy have been proposed. Among them, anti-PD-1 antibody has been newly developed as a next-generation immunotherapy agent that blocks the immune checkpoint pathway[6, 7]. Notably, PD-1 therapy provides significant clinical benefits for patients with an advanced stage cancer[8–11]. Despite this therapy having a substantial effect in advanced cancers in many cases, 20–40% of patients still show progressive disease. A recent clinical report described that, in advanced PD-L1-positive non-small-cell lung cancers, anti-PD-1 therapy as a first-line treatment is associated with significantly longer progression-free survival and improved overall survival compared with the standard platinum-based chemotherapy[12]. In addition, high responders frequently show elevated PD-L1 expression due to gene amplification and upregulation of an ectopic promoter by translocation[13, 14]. Thus, the evidence suggests that PD-L1 level in tumours is an important factor influencing therapeutic efficacy for responders[15–17], although the correlation is not perfect and the mechanisms underlying this correlation in non-responders have not been precisely revealed[18]. Recent studies have suggested that cancers with mutations of genes contributing to genomic stability could be targets for anti-PD-1 therapy. Significantly high rates of responses in cancer with microsatellite instability (MSI), which is a hallmark of genome instability, to anti-PD-1 therapy have been reported in colon cancer[19, 20]. Accumulating studies suggested that MSI-positive tumours presenting neoantigens promote the release of IFNγ from tumour-infiltrating lymphocytes (TILs), and the released IFNγ upregulates PD-L1 expression in immune cells and tumours[21–24]. Therefore, PD-1 therapy is considered to be useful for tumours exhibiting high MSI. Consistent with this notion, patients with mismatch-repair (MMR)-deficient cancer showed a higher rate of progression-free survival following anti-PD-1 therapy, demonstrating that MMR status is a potent predictive marker[25]. High expression of PD-1/PD-L1 in TILs was also observed in DNA polymerase ε-mutated and MSI cancers[26]. Furthermore, the possible involvement of homologous recombination (HR) repair has been suggested recently; specifically,

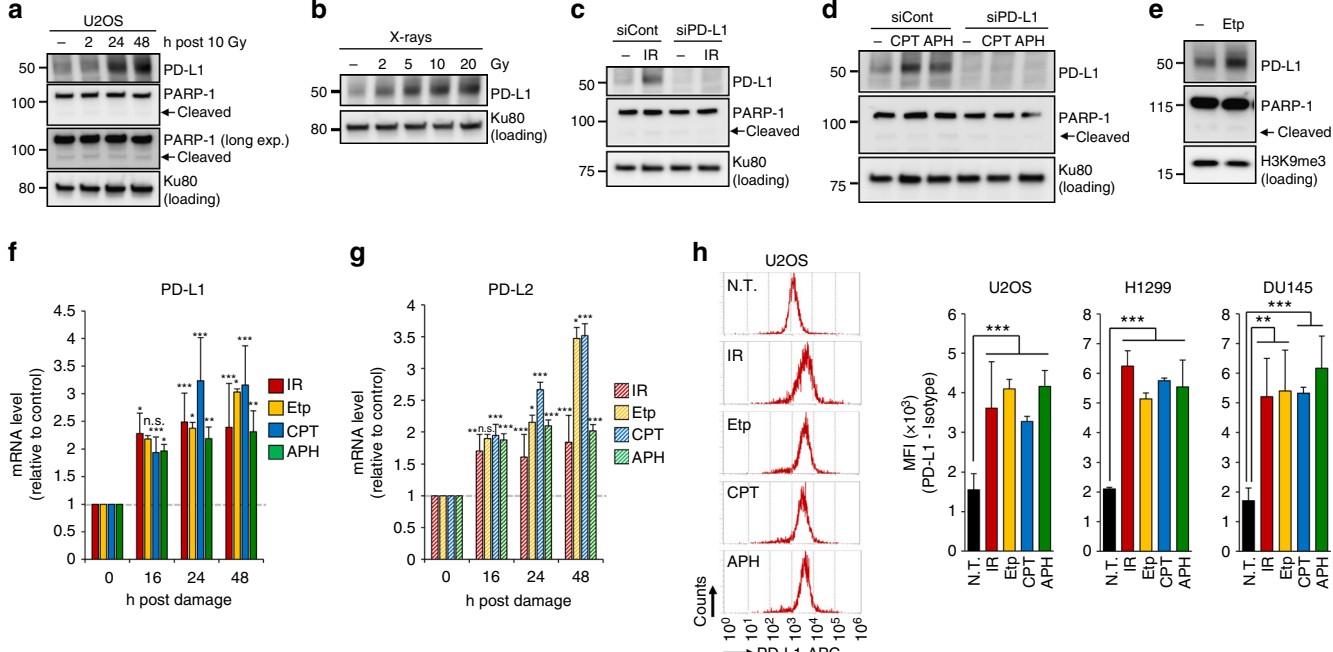

**Fig. 1** DNA damage upregulates PD-L1 expression in cancer cells. **a** PD-L1 was upregulated after IR. PD-L1 expression in U2OS cells was examined 2, 24 and 48 h after 10 Gy. No substantial PARP-1 cleavage was observed during the analysis, suggesting that PD-L1 upregulation was not caused by apoptosis. **b** PD-L1 was upregulated in an IR-dose dependent manner. PD-L1 in U2OS cells was examined at 48 h after 2, 5, 10 and 20 Gy. **c** Depletion of PD-L1 by siRNA in U2OS cells verified that the signal detected at 48 h after 10 Gy X-rays was PD-L1. **d** PD-L1 was upregulated after exposure to DNA-damaging agents. U2OS cells were treated with 50 nM CPT or 500 nM APH. PD-L1 was examined 48 h after the treatment. **e** PD-L1 was upregulated after exposure to Etp. U2OS cells were treated with 500 nM Etp. PD-L1 was examined 48 h after treatment. **f, g** DNA damage upregulates PD-L1 and PD-L2 mRNA expression. U2OS cells were irradiated at 10 Gy, or were treated with 500 nM Etp, 50 nM CPT or 500 nM APH. PD-L1 or PD-L2 mRNA was examined at the indicated time points. Statistical significance was examined compared with non-treated cells. **h** Cell-surface expression of PD-L1 was examined by using flow cytometry. Cell-surface PD-L1 in U2OS, H1299 and DU145 cells was measured 48 h after 10 Gy, 500 nM Etp, 50 nM CPT or 500 nM APH treatment. A representative histogram is shown in left panel. N.T. non-treatment. Error bars represent the s.d. of three independent experiments (**f–h**). Statistical significance was determined using Student's two-tailed t-test. *P < 0.05, **P < 0.01, ***P < 0.001

enrichment of mutations in BRCA2, an HR factor, has been identified in melanomas responsive to anti-PD-1 therapy[27]. HR deficient tumours were also shown to exhibit greater neoantigen loads, TILs and PD-1/PD-L1 expression in immune cells[23]. Thus, the evidence suggests that genomic instability causes high levels of mutations and neoantigen loads in tumours, resulting in greater PD-1/PD-L1 expression in cells surrounding tumour microenvironment.

In cancer treatment, radiotherapy and chemotherapy (e.g. using platinum drugs and alkylating agents) are applied to induce lethal DNA damage in cancer cells. Recent studies have shown that ionising radiation (IR) synergistically promotes antitumor immunity when applied in combination with immune checkpoint inhibitors[28, 29]. Importantly, PD-L1 expression in cancer cells was found to be transiently upregulated following IR, that is, for several days after irradiation, and when the PD-1/PD-L1 interaction was blocked during this upregulation, the anti-PD-1 antibody treatment rescued T-cell activity and delayed tumour growth after irradiation in an immunocompetent mouse model[30, 31]. Anti-PD-1 therapy was also found to enhance the efficacy in patients treated with carboplatin, suggesting its beneficial effect when used in combination with chemotherapy[32]. DNA double-strand breaks (DSBs) are the most critical type of DNA damage induced by radio/chemotherapy and activate signal transduction by phosphoinositol-3-kinase-related kinases, ataxia telangiectasia mutated (ATM), ataxia telangiectasia and Rad3-related protein (ATR) and DNA-PKcs. Such signalling was previously shown to promote DNA repair and arrest the cell cycle, or to trigger apoptosis. In addition to these functions, previous studies have shown that induction of DNA damage response upregulates the expression of cell-surface molecules, such as NKG2D ligand[33]. However, so far, the relationship between DSB-dependent damage signal and PD-L1 expression in cancer cells has not been elucidated. In this study, we showed for the first time that DSBs upregulate PD-L1 expression in an ATM/ATR/Chk1-dependent manner. Depletion of BRCA2 enhances PD-L1 upregulation after DSBs. PD-L1 expression in Ku-depleted cells requires Chk1 after Exonuclease1 (EXO1)-dependent resection. Finally, we showed that DSBs activate STAT1/STAT3 phosphorylation and IRF1, and IRF1 is required for DSB-dependent PD-L1 upregulation, i.e. through the canonical pathway. This new understanding of the molecular mechanism underlying PD-L1 expression in cancer cells should aid the development of strategies to predict prognosis and may contribute to improving the efficacy of therapies in DNA-repair-defective cancer cells, particularly when anti-PD-1 therapy and radio/chemotherapy are combined.

## Results

**DNA damage upregulates PD-L1 expression in cancer cells**. To test the responsiveness of PD-L1 expression in cancer cells after DNA damage, we examined PD-L1 protein levels by immunoblot analysis. PD-L1 expression in osteosarcoma (U2OS), lung cancer (H1299, A549) and prostate cancer (DU145) cell lines was upregulated after IR, which induces DSBs, in a time- and dose-dependent manner (Fig. 1a, b and Supplementary Fig. 1a–d). The upregulation of PD-L1 was observed without substantial PARP-1 cleavage, which is a marker of apoptosis, at 24–48 h after IR (Fig. 1a). An irradiating dose of 10 Gy was toxic and severely damaged the cells; however, cell death was not observed by detecting PARP-1 cleavage in this time range (Fig. 1a; see also other Figures). γH2AX foci analysis and colony formation assay were carried out to examine DSBs levels and cell viability after DNA damage (Supplementary Fig. 2a–c). The IR-dependent upregulation of PD-L1 was not maintained for more than 14 days after IR (Supplementary Fig. 2d). In contrast, paclitaxel, a non-

DNA damaging chemotherapeutic agents, did not cause PD-L1 upregulation (Supplementary Fig. 2e–g).

U2OS cells were mainly used in this study because they have been widely utilised in the analysis of DNA repair and signalling due to their intact DNA damage response compared with that of other cancer cell lines. Thus, these findings suggest that the upregulation of PD-L1 occurs in living cancer cells after IR. In contrast, we did not observe an obvious PD-L1 upregulation in primary fibroblast cells (48BR) after DNA damage in the time range of our analysis (Supplementary Fig. 3a). To verify the antibody specificity against PD-L1 in the immunoblot analysis, we carried out an experiment using PD-L1 siRNA. The PD-L1 signal disappeared due to exposure to PD-L1 siRNA, confirming that the IR-induced signal represents PD-L1 (Fig. 1c). Next, to address the issue of whether PD-L1 expression is dependent on the DNA damage signal or is an IR-specific response, cells were treated with camptothecin (CPT; a topoisomerase I inhibitor, which causes single-strand breaks and forms replication-associated DSBs in S phase cells), aphidicolin (APH; a DNA polymerase inhibitor, which causes DNA replication-associated damage) or Etoposide (Etp; a topoisomerase II inhibitor, which directly causes DSBs). Similar to the upregulation after IR, a significant increase in PD-L1 expression was observed after the treatment with CPT, APH or Etp (Fig. 1d, e). Furthermore, treatment with DNA-damaging chemotherapeutic agents, cisplatin (CDDP), mitomycin C (MMC) and temozolomide (TMZ), also upregulated PD-L1 expression (Supplementary Fig. 3b–d). To confirm the increase in PD-L1 protein after DNA damage is upregulated at the transcriptional level, mRNA was quantified by real-time PCR. Notably, the mRNA of PD-L1 was upregulated 16–48 h after IR, Etp, CPT or APH treatment (Fig. 1f; the time-dependent increase in PD-L1 mRNA after IR is shown in Supplementary Fig. 3e). An increase in PD-L2 expression was also observed after DNA damage (Fig. 1g). For further examination of PD-L1 expression on the cell surface, we measured the mean fluorescence intensity of PD-L1 by flow cytometry (Fig. 1h). Consistent with the above findings, the upregulation of cell-surface PD-L1 in PI-negative cells (live cells) was observed in U2OS, H1299 and DU145 cell lines after IR, Etp, CPT or APH treatments (Fig. 1h). The increase in cell-surface expression after IR was also observed by immunofluorescence analysis (Supplementary Fig. 3f, g). The presence of IFNγ in the culture medium increased PD-L1 expression, and further IR induced the enhancement of PD-L1 expression, although the enhancement may not be synergistic (Supplementary Fig. 4a, b).

Taken together, these findings show that the expression of PD-L1 is upregulated in response to DNA damage, particularly DSBs, in living cancer cells. This suggests that PD-L1 expression in cancer cells is regulated by DNA damage signalling.

**PD-L1 upregulation requires ATM/ATR/Chk1 activity after DSBs**. After IR, DSBs are the most critical type of DNA damage and activate the DNA damage signalling. Etp directly induces DSBs, and other DNA-damaging agents (CPT, APH, CDDP, MMC and TMZ) also induce DSBs but indirectly via DNA replication. ATM, which is the most important signal transducer, serves as a sensor of DSBs[34]. Activated ATM promotes DSB repair as well as cell cycle checkpoint arrest and transduces the signals to downstream effectors in response to DSBs. To determine whether the upregulation of PD-L1 requires ATM kinase activity, we examined PD-L1 expression after IR, CPT, Etp or APH in the presence of a specific ATM inhibitor (ATMi). Strikingly, IR-, CPT-, Etp- or APH-dependent induction of PD-L1 expression was significantly suppressed by ATM inhibition, suggesting that PD-L1 expression is upregulated in response to

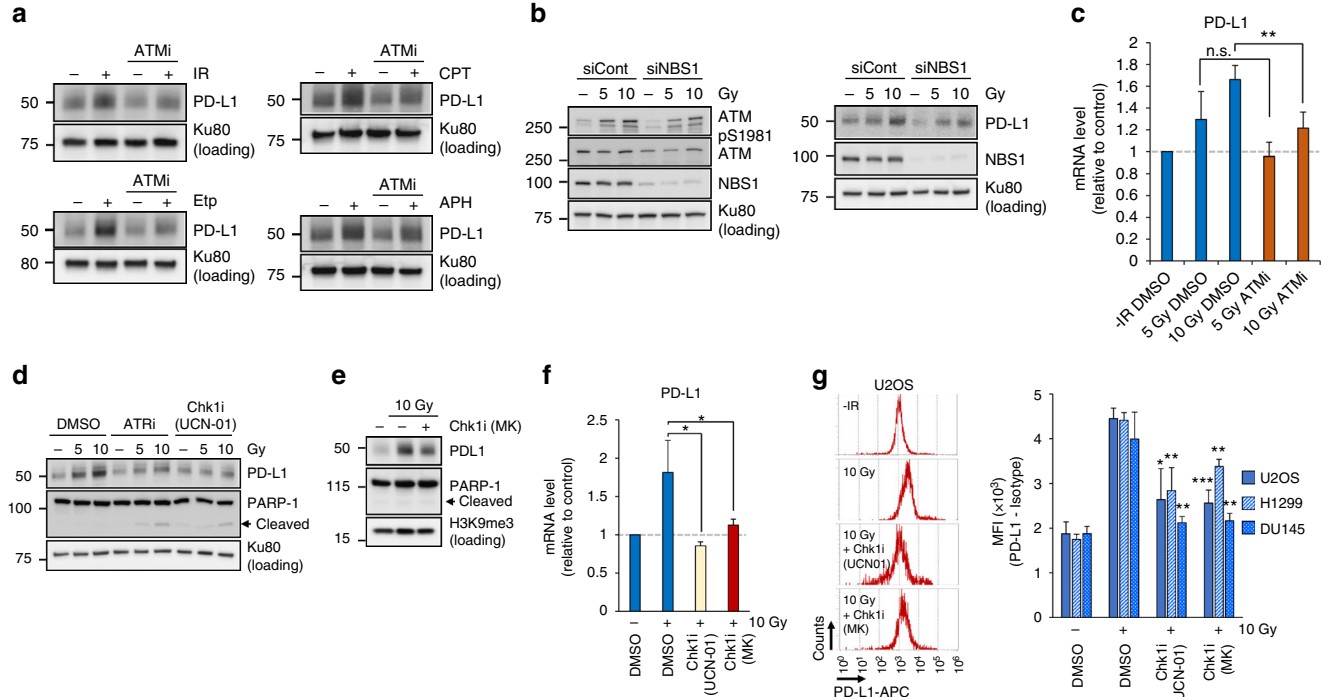

**Fig. 2** PD-L1 upregulation requires ATM/ATR/Chk1 activity after DSBs. **a** ATM kinase activity is required for PD-L1 upregulation after IR. U2OS cells were treated with a specific ATMi 15 min prior to IR, Etp, CPT or APH treatment. PD-L1 was examined 48 h after 10 Gy or the addition of 500 nM Etp, 50 nM CPT or 500 nM APH. **b** NBS1 depletion attenuates ATM auto-phosphorylation, which results in less PD-L1 upregulation. U2OS cells were exposed to NBS1 siRNA. ATM S1981 auto-phosphorylation was examined 30 min after 5 or 10 Gy (left panel). PD-L1 was examined 48 h after 5 or 10 Gy (right panel). **c** ATM kinase activity is required for the upregulation of PD-L1 mRNA expression. U2OS cells were harvested with or without ATMi after IR. PD-L1 mRNA was examined 48 h after 5 or 10 Gy. **d** ATR and Chk1 kinase activities are required for PD-L1 upregulation after IR. PD-L1 in U2OS cells was examined 48 h after 5 or 10 Gy with or without ATRi or Chk1i (UCN-01). **e** The Chk1 dependence was confirmed by another Chk1 inhibitor. PD-L1 in U2OS cells was examined 48 h after 10 Gy with or without Chk1i (MK8776). **f** Chk1 kinase activity is required for the upregulation of PD-L1 mRNA expression. U2OS cells were exposed to IR with or without Chk1i (UCN-01 or MK8776). PD-L1 mRNA was examined 48 h after 10 Gy. **g** Chk1 kinase activity is required for the upregulation of cell-surface PD-L1 after IR. PD-L1 in U2OS, H1299, DU145 cells was examined by flow cytometry at 48 h after 10 Gy with or without Chk1i (UCN-01 or MK8776). Statistical significance was examined compared with non-treated cells. Error bars represent the s.d. of three independent experiments (**c**, **f**, **g**). Statistical significance was determined using Student's two-tailed *t*-test. *$P < 0.05$, **$P < 0.01$, ***$P < 0.001$

DSB-dependent damage signalling (Fig. 2a). The MRN (MRE11/RAD50/NBS1) complex plays a role in amplifying ATM activity; therefore, the lack of NBS1 results in failure to activate ATM signalling[35]. To confirm the finding that activation of ATM is required for PD-L1 upregulation in response to DSBs, we examined PD-L1 in cells exposed to NBS1 siRNA. As predicted, PD-L1 upregulation was reduced in NBS1-depleted cells, confirming that activation of ATM is required for PD-L1 upregulation after DSBs (Fig. 2b; the reduction of ATM activation in NBS1 siRNA cells was confirmed by detecting ATM Ser1981 autophosphorylation). The expression of PD-L1 mRNA was also suppressed by ATM inhibition (Fig. 2c). After IR, ATM is immediately activated at DSB sites. Following the transient activation of ATM, the progression of DSB repair promotes a switch in a signal kinase from ATM to ATR, followed by Chk1 activation[36]. Therefore, to investigate whether PD-L1 upregulation requires downstream kinases, namely ATR and Chk1, PD-L1 levels after IR were examined in the presence of a specific ATR or Chk1 inhibitor (ATRi or Chk1i). Notably, IR-induced PD-L1 upregulation was substantially suppressed by ATR or Chk1 inhibition (Fig. 2d). The PD-L1 upregulation was also suppressed by another Chk1 inhibitor (MK8776) (Fig. 2e). Furthermore, PD-L1 mRNA upregulation was significantly reduced by Chk1 inhibition (Fig. 2f). The upregulation of cell-surface PD-L1 in U2OS, H1299 and DU145 cells was significantly reduced by Chk1 inhibition (Fig. 2g). Taken together, these findings showed that

PD-L1 upregulation after DSBs requires ATM/ATR/Chk1 activity, suggesting that PD-L1 is upregulated in the DSB signal axis. Together, our data strongly suggest that PD-L1 expression is upregulated by DNA damage signalling pathway through the repair in living cells.

**Depletion of BRCA2 enhances PD-L1 upregulation after DSBs.** The findings above prompted us to identify the factors regulating PD-L1 expression in DSB repair and signalling. To identify DSB repair factors regulating PD-L1 upregulation after IR, we carried out a screen using an siRNA library targeting DSB repair and signalling (Supplementary Fig. 5). To identify a factor influencing PD-L1 upregulation following DSBs, PD-L1 expression levels were examined by immunoblot after IR (Fig. 3a, b and Supplementary Fig. 5). Without DNA damage, there was no substantial increase in PD-L1 expression by siRNAs compared with siControl, although the expression exhibited variability among siRNAs (Fig. 3a). Notably, we found that depletion of Ku80, BRCA2 or PALB2 substantially enhanced IR-induced PD-L1 upregulation compared with siControl (Fig. 3b).

First, we focused on BRCA2, a central factor promoting HR in DSB repair (PALB2 has a role in promoting BRCA2 recruitment at DSB sites, so the loss of PALB2 causes a defect in DSB repair in the BRCA2 axis). Consistent with the data in the first siRNA screen, depletion of BRCA2 by using another siRNA targeting BRCA2 substantially increased PD-L1 expression after IR

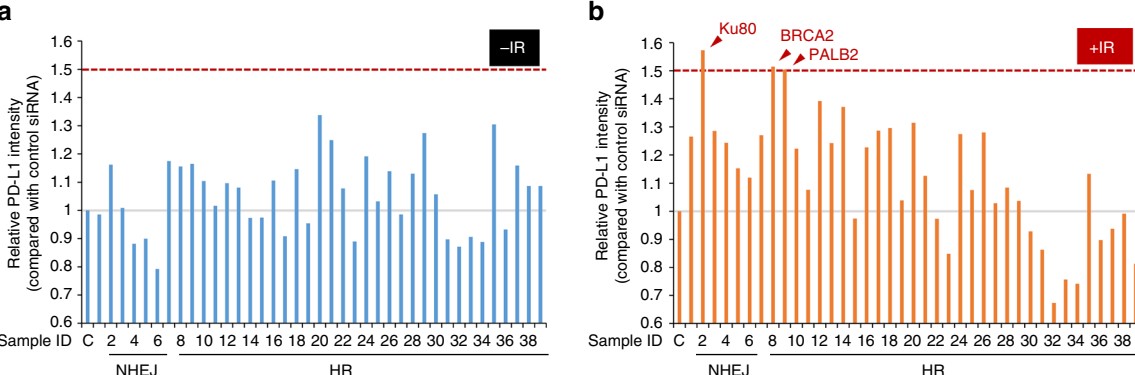

**Fig. 3** Identification of factors influencing PD-L1 upregulation after IR. **a**, **b** PD-L1 intensity relative to that of control siRNA cells was examined after IR by immunoblotting. A raw image of the immunoblotting is shown in Supplementary Fig. 5. U2OS cells were exposed to a Dharmacon siRNA pool containing four distinct oligonucleotides. PD-L1 was examined 48 h after 10 Gy

(Fig. 4a). In a clinical trial, a PARP inhibitor (PARPi) has been utilised to induce synthetic lethality of cancer cells defective in the BRCA2 gene because it blocks single-strand break repair, resulting in the formation of cytotoxic DSBs[37]. Therefore, we tested whether PARP inhibition causes an increase in PD-L1 expression in BRCA2-depleted cells. Notably, the continuous treatment with PARPi significantly upregulated PD-L1 expression in BRCA2-depleted cells (Fig. 4b and Supplementary Fig. 6a). The enhanced PD-L1 expression following PARP inhibition was also significantly suppressed by Chk1 inhibition (Fig. 4c and Supplementary Fig. 6b). Furthermore, an enhancement of PD-L1 mRNA expression in BRCA2-depleted cells after IR was suppressed by ATMi or Chk1i (Fig. 4d, e); similarly, enhancement of PD-L1 mRNA expression after IR was suppressed by Chk1 inhibition in H1299 cells (Supplementary Fig. 6c). In addition, cell-surface upregulation of PD-L1 after IR or PARPi was further enhanced by BRCA2 depletion in U2OS and H1299 cells (Fig. 4f, g and Supplementary Fig. 7). Notably, the enhancement of PD-L1 cell-surface expression was suppressed by Chk1 inhibition (Fig. 4h–k and Supplementary Fig. 7), demonstrating that the upregulation of PD-L1 in BRCA2-depeleted cells requires Chk1 activity. Together, these results demonstrate that BRCA2 is a critical factor regulating PD-L1 expression after DSBs.

**Depletion of Ku complex enhances PD-L1 expression after DSBs.** Alongside the finding of a greater increase of PD-L1 upregulation in BRCA2-depleted cells, we also found a substantial increase in PD-L1 by Ku80 siRNA after IR. Similarly, Ku70 siRNA increased PD-L1 upregulation after IR (Fig. 5a). Consistent with previous reports, the loss of either Ku70 or Ku80 led to disruption of the Ku70/80 complex (Fig. 5a). This experiment suggests that the increase in PD-L1 is not due to an off-target effect of siRNA. It has been shown that the Ku70/80 complex is a critical factor for non-homologous end joining (NHEJ), which is the major pathway of DSB repair in human cells[38]. The Ku70/80 complex has two major roles in DSB repair. One is the recruitment of NHEJ factors to initiate the down-stream part of the repair process. The other is to protect DNA ends from unscheduled digestion by DNA nucleases[39, 40]. Since DSB-dependent PD-L1 expression requires Chk1 activity, which is activated by ATR on single-stranded DNA (ssDNA)-replication protein A (RPA), we hypothesised that the greater PD-L1 upregulation after IR in Ku80-depleted cells may be due to Chk1 overactivation following an increase in DNA end resection. To address this issue, we examined resection activity and Chk1 signal by monitoring the phosphorylation of RPA Ser4/8 (a resection marker) and Chk1 Ser345, respectively. Greater RPA/Chk1

phosphorylation was observed in Ku80-depleted cells after IR, suggesting that Ku80 depletion enhances DNA end resection and Chk1 activation (Fig. 5b). To further test whether the increases in resection and Chk1 activity require DNA nucleases, we conducted an experiment using siRNA of EXO1, which is a major DNA exonuclease in HR repair, and examined RPA/Chk1 phosphorylation after IR in cells subjected to EXO1/BLM siRNA in a Ku80-depleted background (N.B. depletion of BLM, a DNA helicase, is required because it cooperatively promotes EXO1-dependent resection in HR)[41]. Notably, EXO1/BLM depletion significantly reduced RPA/Chk1 phosphorylation in Ku80-depleted cells after IR, suggesting that EXO1/BLM is required for resection and Chk1 activation in a Ku80-depleted background (Fig. 5b). Next, we examined whether PD-L1 upregulation in Ku80-depleted cells is dependent on EXO1/BLM. Consistent with the reduction of RPA/Chk1 phosphorylation, the depletion of EXO1/BLM cancelled the upregulation of IR-induced PD-L1 expression in Ku80-depleted cells (Fig. 5c). To analyse further whether the upregulation of PD-L1 in Ku80-depleted cells requires Chk1 activity, we examined PD-L1 expression in cells treated with Chk1i. Chk1 inhibition significantly reduced PD-L1 upregulation in Ku80-depleted cells (Fig. 5d). Next, to consolidate that PD-L1 upregulation in Ku80-depleted cells is dependent on DSB-induced damage signalling, we examined PD-L1 expression after Etp treatment. Similar to the results after IR, Ku80 depletion caused further upregulation of PD-L1 after Etp treatment (Fig. 5e). The increase in DSB end resection by Ku80 depletion was confirmed by the greater number of RPA foci after Etp (Fig. 5f). Notably, the inhibition of ATM or Chk1 activity also cancelled the upregulation of PD-L1 mRNA (Fig. 5g). Furthermore, an enhancement of PD-L1 cell-surface expression by Ku80 depletion was observed, and it was supressed by Chk1 inhibition (Fig. 5h–j and Supplementary Fig. 8). Taken together, these results suggest that loss of the Ku70/80 complex upregulates PD-L1 via Chk1 activation following the increase in EXO1/BLM-dependent resection after IR.

**PD-L1 upregulation after DSBs is mediated via IRF1 pathway.** A recent report demonstrated that the JAKs–STATs–IRF1 pathway primarily regulates PD-L1 expression after treatment with IFNγ[42]. We, therefore, examined whether DSBs induce the activation of STAT1, STAT3 and IRF1 pathways. We found that IRF1 and STAT1/3 phosphorylation was upregulated in response to DSBs (IR, Etp or CPT) in U2OS and H1299 cells (Fig. 6a, b). Notably, the depletion of IRF1 significantly reduced PD-L1 upregulation after IR, suggesting that DSB-dependent PD-L1 upregulation is mediated by IRF1 pathway (Fig. 6c). Also, we

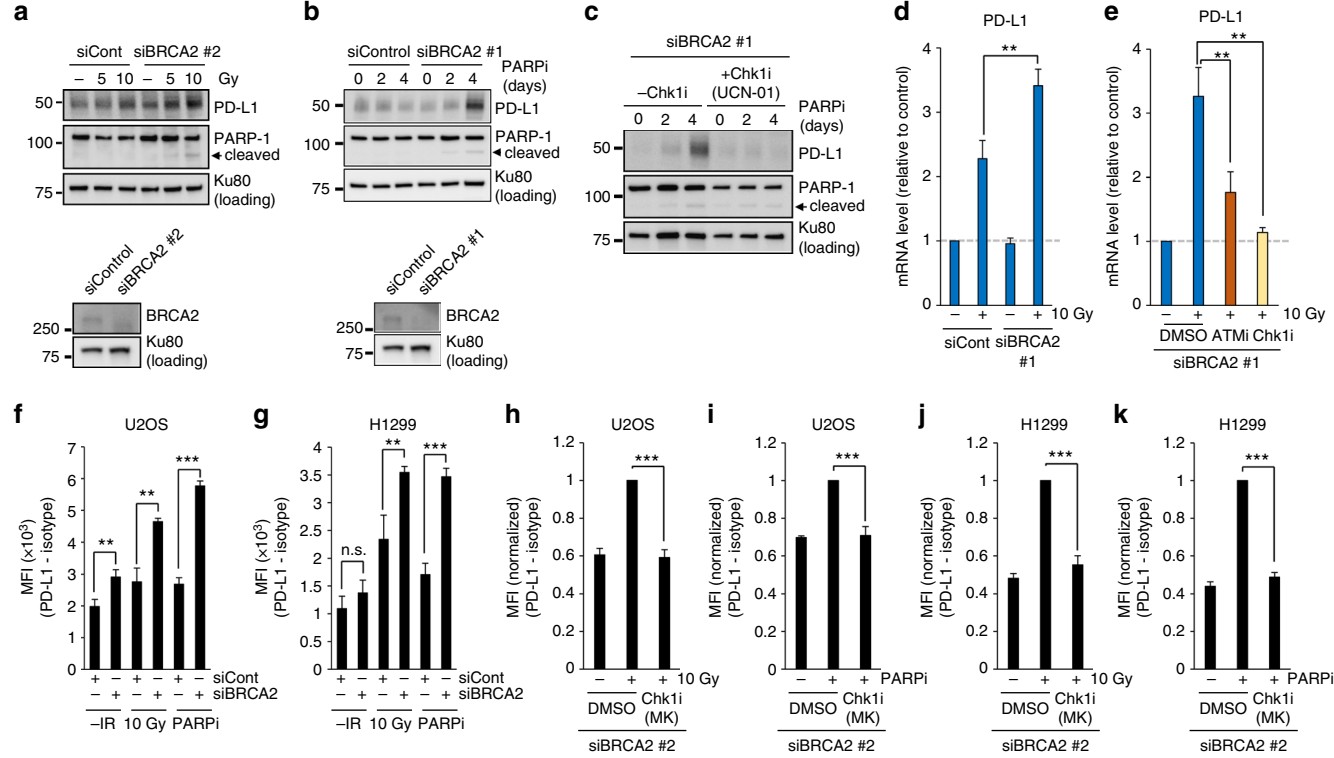

**Fig. 4** Depletion of BRCA2 enhances PD-L1 upregulation after DSBs. **a** Depletion of BRCA2 enhances the upregulation of PD-L1 after IR. U2OS cells were exposed to BRCA2 siRNA. PD-L1 was examined 48 h after 5 or 10 Gy. Knockdown efficiency of BRCA2 is shown in the bottom panel. **b** PARP inhibition enhanced PD-L1 upregulation in BRCA2-depleted cells. U2OS cells were exposed to BRCA2 siRNA. PD-L1 was examined 2 and 4 days after the addition of PARPi. Knockdown efficiency of BRCA2 is shown in the bottom panel. Similar results were obtained using a distinct siRNA (Supplementary Fig. 6a). **c** The enhancement of PD-L1 expression in BRCA2-depleted cells following PARP inhibition requires Chk1 activity. PD-L1 in U2OS cells was examined 2 and 4 days after the addition of PARPi with or without Chk1i (UCN-01). Similar results were obtained using a distinct siRNA (Supplementary Fig. 6b). **d** Depletion of BRCA2 enhanced the upregulation of PD-L1 mRNA after IR. PD-L1 mRNA in U2OS cells with or without BRCA2 siRNA was examined 48 h after 10 Gy. **e** ATM/Chk1 activity is required for the upregulation of PD-L1 mRNA in BRCA2-depleted cells after IR. PD-L1 mRNA in U2OS cells following BRCA2 siRNA with or without ATMi or Chk1i (UCN-01) was examined 48 h after 10 Gy. Similar results were obtained in H1299 cells (Supplementary Fig. 6c). **f**, **g** Depletion of BRCA2 enhanced the upregulation of cell-surface PD-L1 after IR or PARP inhibition. Cell-surface PD-L1 expression in U2OS (**f**) or H1299 (**g**) cells following BRCA2 depletion was examined by flow cytometry 48 h after 10 Gy. **h–k** The upregulation of cell-surface PD-L1 requires Chk1 activity. Cell-surface PD-L1 expression in U2OS cells (**h**, **i**) was examined with or without Chk1 inhibitor after IR or PARPi. Cell-surface PD-L1 expression in H1299 cells was examined with or without Chk1 inhibitor after IR or PARPi (**j**, **k**). Error bars represent the s.d. of three independent experiments (**d–k**). Statistical significance was determined using Student's two-tailed t-test. *P < 0.05, **P < 0.01, ***P < 0.001

found that STAT1 Try701 was not effectively phosphorylated in IRF1-depleted cells after IR. This may suggest that there is a feedback from IRF1 to STAT1 phosphorylation. Next, to address the involvement of ATM/Chk1 signalling in the IRF1 activation, IRF1 levels were examined with or without ATMi or Chk1i after IR. Inhibition of ATM or Chk1 activity reduced IRF1 expression after IR (Fig. 6d, e). Further, PD-L1 cell-surface upregulation after IR was diminished by IRF1 siRNA (Fig. 6f, g). Together, these results suggest that DSB-dependent signalling upregulates PD-L1 expression, and importantly, this upregulation requires the activation of STAT1/3–IRF1 pathway.

Finally, we investigated the PD-L1 levels in neoplastic samples based on the data set from The Cancer Genome Atlas (TCGA). We found that PD-L1 in tumours (breast, stomach, colorectal and uterine cancers) with BRCA2, PALB2 or Ku70/80 mutations showed higher expression than in tumours without such mutations (Supplementary Fig. 9a–c). All the selected samples exhibiting mutations in any of these genes showed significantly higher PD-L1 expression (Supplementary Fig. 9d; the analyses of other tumours are shown in Supplementary Table 1). Tumours with mutations in Chk1 showed the tendency of lower PD-L1 expression, although there was no statistical significance, possibly due to the small number of Chk1 mutated samples

(Supplementary Fig. 9e). In contrast, tumours with mutation with the p53 gene did not show an increase in PD-L1 expression (Supplementary Fig. 10). The levels of neoantigens in neoplastic samples harbouring BRCA2, PALB2 or Ku70/80 mutations were also examined in TCGA. A significant increase in neoantigens was observed in breast and colorectal samples, whereas no statistical significance was shown in stomach and uterine samples (Supplementary Fig. 11).

## Discussion

Mounting evidence suggests that the efficacy of PD-1 therapy is related to genomic instability, including MSI, due to MMR and DNA replication stress[19, 20, 25, 26]. The efficacy of PD-1 therapy in cancer cells showing genomic instability or lacking the capacity to perform DNA repair is explained by the high production of neoantigen in cancer cells. In addition, recent studies proposed that high mutational loads in cancer cells cause the release of neoantigen, which activates the signal cascade to recruit TILs and results in PD-1/PD-L1 expression in immune and cancer cells[22, 23, 27]. In contrast, the mechanism behind PD-L1 upregulation by DNA damage in the context of DNA repair and signalling has not been fully investigated, although several studies suggested that the

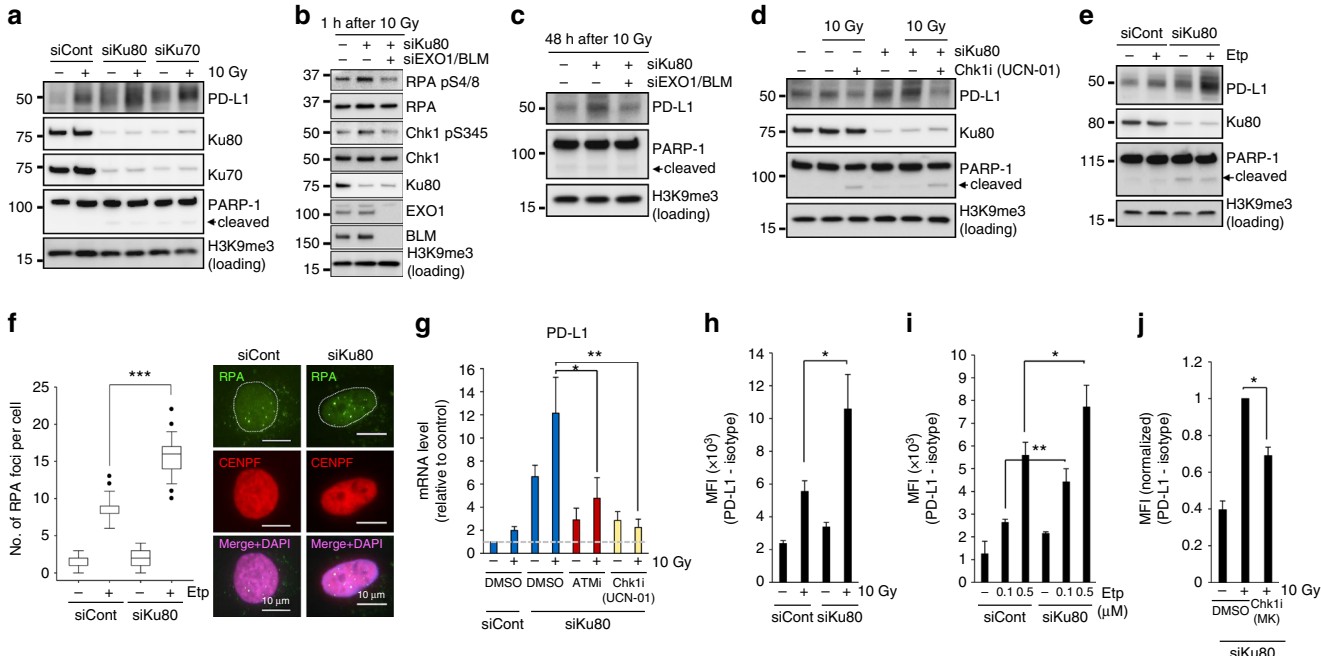

**Fig. 5** Depletion of Ku complex enhances PD-L1 expression after DSBs. **a** Depletion of either Ku80 or Ku70 substantially enhances PD-L1 upregulation after IR. U2OS cells were exposed to Ku80 or Ku70 siRNA. PD-L1 level was examined 48 h after 10 Gy. **b** EXO1/BLM is required for resection, which is determined by phosphorylation of RPA S4/8, in Ku80-depleted cells. The enhancement of Chk1 phosphorylation in Ku80-depleted cells requires EXO1/BLM. U2OS cells were exposed to Ku80 siRNA with or without EXO1/BLM siRNA. The phosphorylation of Chk1 S345 and RPA S4/8 (a marker of resection) was examined 1 h after 10 Gy. **c** The enhancement of PD-L1 in Ku80-depleted cells requires EXO1/BLM. U2OS cells were exposed to Ku80 siRNA with or without EXO1/BLM siRNA. PD-L1 was examined 48 h after 10 Gy. **d** The enhancement of PD-L1 in Ku80-depleted cells requires Chk1 activity. U2OS cells were exposed to Ku80 siRNA. PD-L1 expression was examined in Ku80 siRNA cells with or without Chk1i (UCN-01) 48 h after 10 Gy. **e** Depletion of Ku80 substantially enhances PD-L1 upregulation after Etp treatment. PD-L1 level was examined 48 h after 100 nM Etp treatment. **f** The enhancement of DSB end resection in Ku80-depleted cells was shown by the greater number of RPA foci after 100 nM Etp treatment. The number of RPA foci in G2 cells were examined. G2 cells were identified by CENPF[45]. The scale bar represents 10 μm. **g** ATM/Chk1 activity is required for the upregulation of PD-L1 mRNA in Ku80-depleted cells after IR. PD-L1 mRNA in Ku80-depleted U2OS cells with or without ATMi or Chk1i (UCN-01) was examined at 48 h after 10 Gy. **h–j** Depletion of Ku80 enhanced the upregulation of cell-surface PD-L1 after IR (**h**) or Etp (**i**). This upregulation requires Chk1 activity (**j**). Cell-surface PD-L1 expression in Ku80-depleted U2OS cells with or without Chk1i (MK8776) was examined at 48 h after 10 Gy or Etp treatment. Error bars represent the s.d. of three independent experiments (**g–j**). Statistical significance was determined using Student's two-tailed *t*-test. *$P < 0.05$, **$P < 0.01$, ***$P < 0.001$

blockage of PD-1/PD-L1 interaction enhances in vivo tumour growth delay in combination with IR[28, 30, 31]. Here we demonstrate that the repair of DSBs is a factor regulating PD-L1 expression in cancer cells. We discovered that PD-L1 upregulation in cancer cells is ATM/ATR/Chk1-dependent after IR or treatment with CPT, APH or Etp. The requirement for these kinases strongly supports the notion that PD-L1 upregulation is controlled by the DNA damage signalling following DSB induction in living cells. The DNA-damaged cells are gradually dying or cell growth is finally arrested due to cellular senescence within 1–2 weeks. Although >90% of cells did not survive after 10 Gy X-rays, PD-L1 expression levels return back to normal in surviving cells. Such transient upregulation of PD-L1 in DNA-damaged cells might be induced for preventing overactivation of immune activity surrounding the tumours. We further show that the upregulation of PD-L1 is enhanced when a specific DSB repair protein, BRCA2 or Ku70/80, is depleted. We stress that the repair of DSBs per se is not a factor regulating PD-L1 expression in cancer cells, and we suggest that the activation of Chk1 following DNA end resection is a critical step leading to the upregulation of PD-L1 (Fig. 7).

DSBs are repaired by two major pathways: NHEJ and HR. NHEJ occurs throughout the cell cycle in mammalian cells, while HR repairs DSBs in the S/G2 phase of the cell cycle in a CDK-dependent manner[38]. Despite the pro-HR environment in the

S/G2 phase, current models suggest that Ku70/80 heterodimers bind rapidly to DSBs, allowing NHEJ to make the first attempt towards repair[43]. However, if NHEJ does not ensue, the repair pathway can be switched towards HR, which is triggered by MRE11/CtIP endonuclease activity, and this switch is stimulated by BRCA1-dependent RIF1 release[44–46]. In the second step of resection, exonucleases such as EXO1/BLM expand resection by digesting DNA to produce a sufficient length of ssDNA[41, 44]. Following DNA end resection, ssDNA is coated by RPA, which is replaced by RAD51 to facilitate homology searching and the subsequent steps of HR. ATM is activated at two-ended DSBs at an early stage after IR, namely, before resection, while ATR is activated following RPA recruitment on ssDNA. Because ATM is not activated at ssDNA and ATR is not activated at unresected DSB ends, the progression of resection is required for ATR activation[36]. Subsequently, ATR phosphorylates and activates Chk1. Chk1 is a multi-functional protein, regulating DNA repair, signalling and transcriptional activation. In fact, in the immune response after DNA damage, NKG2D ligands in cancer cells are also upregulated by Chk1 activity[33]. Thus, Chk1 can be an important regulator for inducing a switch from a DNA-damage-related signal to one associated with protein expression, which includes cell surface molecules in immune responses. In addition to the involvement of Chk1 in DSB-dependent PD-L1 upregulation, we discovered that IRF1 is required for the upregulation.

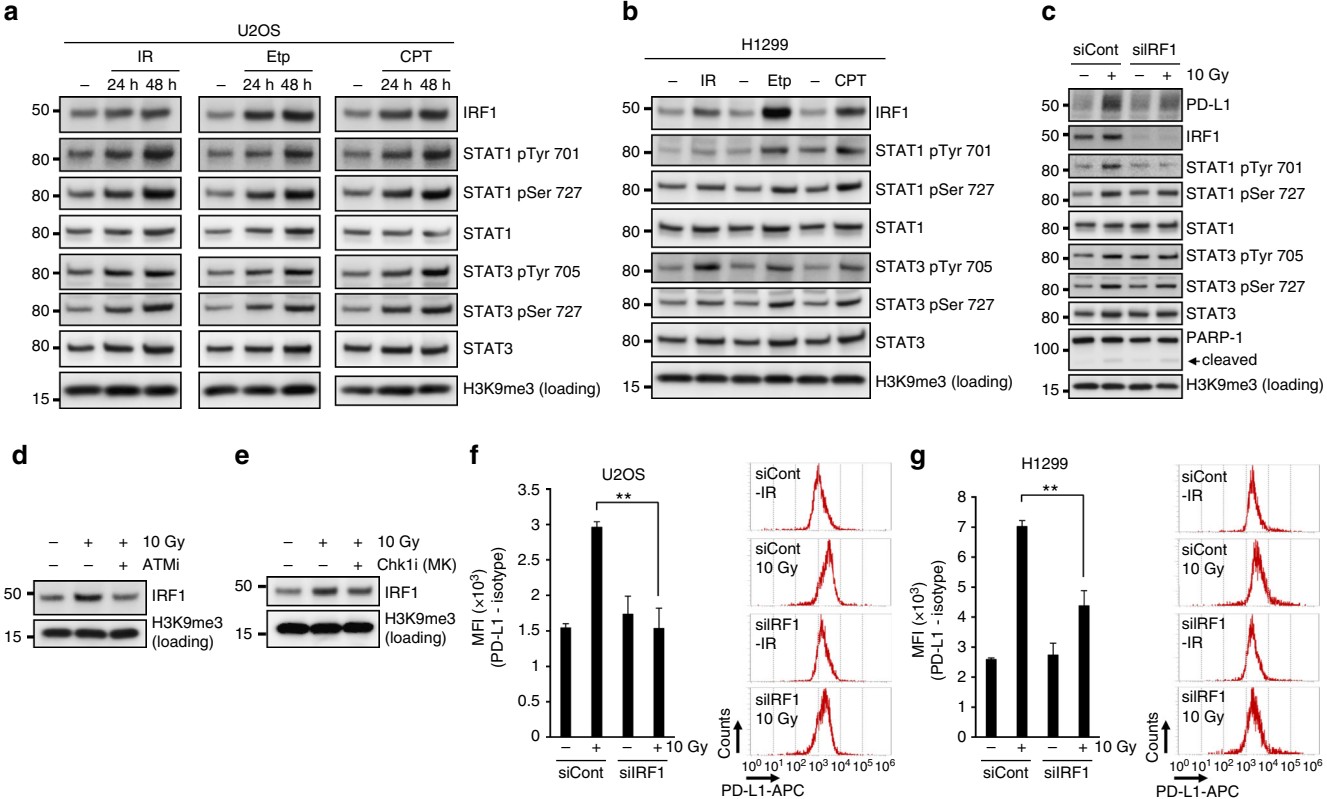

**Fig. 6** DSB-dependent PD-L1 upregulation is mediated through IRF1 pathway. **a**, **b** DSBs activate STAT1/3 and IRF1 signalling. U2OS (**a**) or H1299 (**b**) cells were harvested after 10 Gy, 500 nM Etp or 50 nM CPT. **c** IRF1 is required for PD-L1 upregulation after IR. PD-L1 expression was examined in IRF1-depleted U2OS cells 48 h after 10 Gy. **d**, **e** ATM and Chk1 activities promote IRF1 upregulation after IR. IRF1 levels were examined with or without ATM (**d**) or Chk1 (MK8776) (**e**) inhibitor after 10 Gy. **f**, **g** Depletion of IRF1 significantly reduces the upregulation of cell-surface PD-L1 after IR. Cell-surface PD-L1 expression in IRF1-depleted U2OS (**f**) or H1299 (**g**) cells was examined by flow cytometry at 48 h after 10 Gy. A representative histogram is shown in the right panel. Error bars represent the s.d. of three independent experiments (**f**, **g**). Statistical significance was determined using Student's two-tailed *t*-test. *$P < 0.05$, **$P < 0.01$, ***$P < 0.001$

Previous studies showed that IRF1 is the central factor for PD-L1 induction after IFNγ treatment[42]. The upregulation of PD-L1 by IFNγ is mediated through IRF1 binding to the PD-L1 promoter in the JAK1/2-STAT1/2/3-IRF1 signalling[42, 47, 48]. Notably, increases in STAT1/3 phosphorylation and IRF1 levels were observed following DSBs. Further, depletion of IRF1 significantly reduced PD-L1 upregulation after DSBs. These data suggest that DSB-dependent PD-L1 expression is also regulated via STATs and IRF1 pathway (Fig. 7). However, it is still unclear how Chk1 activates the downstream factors. Further studies are thus required to reveal the role of Chk1 in transcriptional activation of PD-L1 through the network of canonical signal cascades.

A recent cohort study reported associations of CD3+ TILs and BRCA1/2 mutation status with survival[23]. Alternatively, an immunohistochemical analysis in BRCA1-associated and sporadic triple-negative breast cancer (TNBC) showed that there was no significant difference in PD-L1 expression in TNBC between BRCA1 carriers and noncarriers[49]. In our screen, BRCA1-depleted cells did not show a significant increase in IR-induced PD-L1 expression compared with that in control cells. These paradoxical results can be explained by our model that Chk1 activation via DNA end resection is a key process in DSB-induced PD-L1 expression. Both BRCA1 and BRCA2 are required for HR, however, BRCA1 and BRCA2 have distinct roles in HR. BRCA1 promotes DNA end resection by relieving the barrier posed by 53BP1 in HR. Therefore, BRCA1 deficiency attenuates the DNA damage signal after IR unless the 53BP1 pathway is functioning normally[50, 51]. In particular, the ATR/Chk1 signal is not

effectively activated if the resection is impaired in BRCA1-defective cells. Notably, the loss of 53BP1, or the loss of RIF1 or REV7, rescues DNA end resection and HR in BRCA1-defective cells[52–54]. Disruption of the 53BP1 pathway, which rescues BRCA1 defect, was observed in subsets of sporadic TNBC. Notably, ~70% of BRCA-associated TNBC cases were found to revert to HR via an unknown mechanism. Therefore, the BRCA1 gene status alone may not be simply correlated with PD-L1 expression. On the other hand, BRCA2 is dispensable for resection, but is required for RAD51 loading following the formation of RPA on resected DNA ends. The protein switch from RPA to RAD51 is promoted by BRCA2, and the recruitment of BRCA2 on chromatin requires PALB2. Therefore, the defect of BRCA2 causes the stalling of HR repair at the step of RAD51 loading, but sustains activation of Chk1 at unrepaired DSBs. This effect was confirmed when BRCA2-depleted cells were treated with PARPi. PARP inhibition causes replication fork stalling, which requires BRCA2 to resolve and repair DNA. Notably, the addition of a Chk1 inhibitor cancelled the PARPi-dependent PD-L1 upregulation in BRCA2-depleted cells. Hence, we propose that a defect of DSB repair does not simply influence PD-L1 upregulation; however, the Chk1 activity plays a central role in the upregulation of PD-L1 after DSBs.

Ku70/80 is a heterodimer protein that binds to DSB ends and promotes the recruitment of NHEJ proteins[38]. In addition to this role, the Ku complex plays a significant role in protecting DSB ends. To remove the Ku complex from DSB ends safely without excessive resection, the initiation of resection is highly regulated

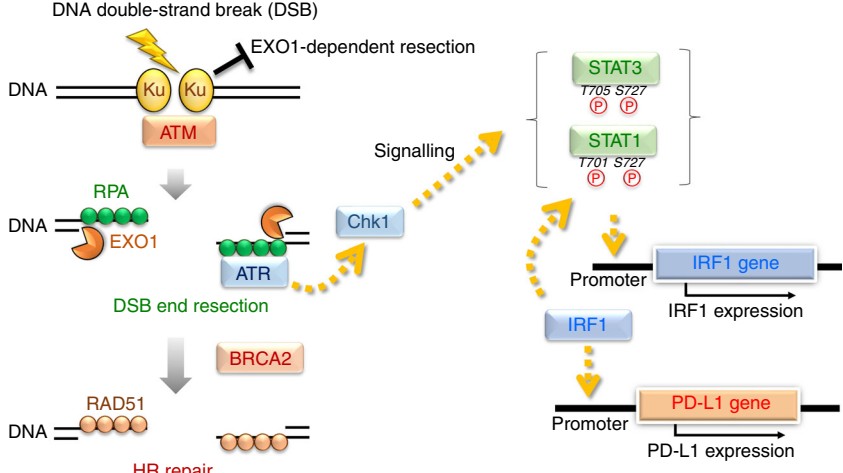

**Fig. 7** Model for PD-L1 upregulation in response to DNA double-strand breaks. In the newly proposed model in the context of DSB repair pathway choice, the Ku70/80 complex binds rapidly to all the DSB ends, allowing NHEJ to make the first attempt at repair immediately after IR[44, 46, 55]. The Ku70/80 complex promotes NHEJ, whereas it prevents unscheduled resection by blocking the access of DNA nucleases. DSBs also activate ATM, a central signal transducer. ATM then activates DNA end resection[45, 55]. ATR/Chk1 is activated onto the resected RPA-coated single-stranded DNA (ssDNA) and at the same time BRCA2 promotes the switch from RPA to RAD51 on ssDNA in the HR pathway. Thus, based on the repair switch model from Ku-binding to HR, our findings suggest a defect in a DSB repair protein, i.e. a defect in either Ku70/80 complex or BRCA2 leads to the upregulation of PD-L1 expression via Chk1 activation following EXO1-dependent resection. Furthermore, we showed that STAT1/3 and IRF1 are activated in response to DSBs. Notably, PD-L1 upregulation requires IRF1, strongly suggesting that DSB-dependent PD-L1 upregulation is induced by the canonical STAT–IRF1 pathway[42]

by MRE11/CtIP endonucleases[44]. However, the lack of the Ku complex causes an increase in DSB end resection, which requires EXO1/BLM[39, 40]. This increase in resection results in Chk1 activation. We found that depletion of the Ku complex caused high level of PD-L1 expression even without DNA damage. The greater endogenous PD-L1 expression (without DNA damage) in Ku80-depleted cells is likely due to the accumulation of replication-associated damage because the Ku complex also binds to replication-associated DSBs[55]. In addition to the upregulation of endogenous PD-L1 expression, we found a substantial increase in PD-L1 by Ku depletion after IR or Etp. An experiment using a Chk1 inhibitor clearly revealed that DSB-induced upregulation of PD-L1 in Ku-depleted cells is mediated by Chk1 following the excessive resection, supporting the notion that DSB-induced PD-L1 is regulated in resection-Chk1 pathway. Throughout the present study, we noticed that the magnitude of PD-L1 upregulation between mRNA and protein levels was not always perfectly matched. Because PD-L1 is controlled by post-translational modifications, such as ubiquitination[56], and also multiple ubiquitin ligases and deubiquitinating enzymes are activated following DSBs[57], the PD-L1 levels may be fine-tuned by the modifications depending on the cellular situation after DNA damage. Further studies are required to reveal the post-translational regulation for controlling PD-L1 expression after DNA damage. In an investigation of clinical specimens, reduction of Ku70/80 expression was observed in samples with the progression of melanoma[58]. In addition, BRCA2 mutations are enriched in melanomas responsive to anti-PD-1. Thus, downregulation of either Ku or BRCA2 or both may affect the efficacy of PD-1 therapy through PD-L1 upregulation by IR, and also via neoantigen pathway.

In previous studies, PD-1 therapy was shown to achieve remarkable clinical responses in patients with several types of cancer; however, ~20–40% of patients still remained poorly responsive[8–11]. In addition, a meta-analysis reported that PD-L1 expression on cancer cells is one of the promising candidates as a selection marker[17]. A phase III clinical trial that involved only patients who had non-small-cell lung cancer with PD-L1 expression on at least 50% of tumour cells showed that

pembrolizumab was associated with significantly longer progression-free and overall survival than chemotherapy[12]. These findings suggest that PD-1 therapy is effective in patients showing high PD-L1 expression. In the TCGA analysis, we showed a correlation between PD-L1 expression and mutations in the BRCA2, PALB2 and Ku70/80 genes. This data set analysis may not represent a direct correlation between PD-L1 expression and DSB signalling because the enhancement might largely be due to the high expression of neoantigens. However, because the statistical significance of the enhancement of neoantigens is less than that of the enhancement of PD-L1, we speculate that DNA damage signalling may partly contribute to the increase in PD-L1 expression in tumours when DNA repair and signalling are downregulated, particularly in patients who are treated with radio/chemotherapy.

Taken together, in the present study, we showed that the depletion of BRCA2 or Ku80 upregulates PD-L1 expression in the DSB signal cascade. Our siRNA screen showed that the downregulation of DSB repair genes is not always associated with PD-L1 upregulation. Notably, we demonstrated that factors regulating Chk1 activity are important. Thus, BRCA2, PALB2 and Ku70/80 identified in this study could be good markers for predicting the efficacy of combined PD-1 and radio/chemotherapy, although a combined therapy may be sufficiently effective regardless of the gene status. In conclusion, we propose that understanding the signalling pathway underlying the response to DSB induction by radio/chemotherapy, which leads to PD-L1 upregulation, is of high importance to develop prognostic molecular markers and PD-1 blockade cancer immunotherapy. The factors could be pharmaceutical targets to influence the therapeutic efficacy when combined PD-1 and radio/chemotherapy is administered. These open questions will be investigated in future work.

## Methods
**Cell culture, irradiation and drug treatment**. U2OS, H1299 and DU145 cells were obtained from the American Type Culture Collection (ATCC). Cancer cells were cultured in Eagle's Minimum Essential Medium (EMEM) with 10% foetal calf serum (FCS). X-ray irradiation was performed at 100 kVp and 20 mA with copper (0.5 mm)–aluminium (1.0 mm) filters (Faxitron Bioptics, Tucson, AZ, USA). Dose rate was set at 0.5 Gy per min. Etp (Sigma-Aldrich), a topoisomerase II inhibitor,

was added at 100 or 500 nM, or CPT (Wako), a topoisomerase I inhibitor, was added at 50 nM, and cells were incubated until harvesting. Etp directly causes DSBs, whereas CPT causes replication-dependent DSBs in S phase cells. APH (Wako), a replicative polymerase inhibitor, was added at 500 nM, and cells were incubated until harvesting. In addition, 10 μM ATM inhibitor (ATMi) (KU55933; Merck Chemicals), 10 μM ATR inhibitor (ATRi) (VE821; Axon Medchem), Chk1 inhibitor (100 nM UCN-01; Calbiochem or 50 nM MK8776; AdooQ Bioscience) was added at 30 min prior to DNA damage induction. A previous study has shown that 100 nM UCN-01 specifically inhibits Chk1 but not Chk2[59]. PARP inhibitor (PARPi) (Olaparib; AdooQ BioScience) was also added at 10 μM, and the medium was refreshed with 10 μM PARPi every 48 h until the indicated time point.

**siRNA knockdown**. siRNA transfection was performed using HiPerFect (Qiagen). siRNA was added to suspended cells after trypsinisation. After 24 h, cells were re-transfected with siRNA in suspended cells after trypsinisation. Cells were incubated for 24 h after the second transfection before DNA damage induction. The sequences of the siRNA oligonucleotides used in this part of the study are listed in Supplementary Table 2.

**Immunoblotting and immunofluorescence**. Immunoblotting was performed by using whole cell lysate[44]. The uncropped versions of immunoblotting are shown in Supplementary Fig. 12. To quantify the cell-surface PD-L1 signal by immuno-fluorescence staining, cells were fixed with 3% paraformaldehyde-2% Sucrose, then the fixed cells were stained with anti-PD-L1 antibody, propidium iodide (PI) and 4′,6-diamidino-2-phenylindole (DAPI) without permeabilisation. Representative images were taken using a Nikon ECLIPSE N*i* microscope by using NIS-Elements D imaging software with a 40 × objective via a DS-Qi2 camera. The mean signal intensity of PD-L1 in DAPI-negative areas was measured by ImageJ v1.48. Cells positive for PI, representing dead cells, were excluded from this analysis. Similar results were obtained in at least two independent experiments. The antibodies used are listed in Supplementary Table 3.

**Quantification of mRNA expression levels by real-time PCR**. Total RNA was extracted from cells using NucleoSpin RNA (MACHEREY-NAGEL) at the indicated time points after DNA damage induction. PrimeScript RT Reagent Kit (Perfect Real Time) (TaKaRa) was used to reverse-transcribe cDNA from total RNA, in accordance with the manufacturer's instructions. Quantitative PCR (qPCR) was performed using StepOnePlus (Life Technologies). Reactions (20 μl each) were prepared in duplicate in MicroAmp Fast Optical 96-Well Reaction Plate (Applied Biosystems). Each reaction contained 0.5 μM of each primer, 0.2 μM probe, 10 μl of Taqman Universal PCR Master Mix (Applied Biosystems) and cDNA as a template. The expression levels were normalised to GAPDH and calculated using the $2^{-\Delta\Delta Ct}$ method. qPCR settings were as follows: initial denaturation at 95 °C for 10 min, followed by 45 cycles of denaturation at 95 °C for 15 s, and annealing and extension at 60 °C for 1 min. The primers and probes used for the qPCR are listed below:

PD-L1 forward: 5′-GGAGATTAGATCCTGAGGAAAACCA-3′
PD-L1 reverse: 5′-AACGGAAGATGAATGTCAGTGCTA-3′
PD-L1 probe: 5′-AGATGGCTCCCAGAATTACCAAGTGAGTCC-3′
PD-L2 forward: 5′-GCTTCACCAGATAGCAGCTTTATTC-3′
PD-L2 reverse: 5′-CTCCAAGGTTCACATGACTTCCA-3′
PD-L2 probe: 5′-CATTCCAGGGTCACATTGCTGCCATGCT-3′
GAPDH forward: 5′-CTCCTCTGACTTCAACAGCGA-3′
GAPDH reverse: 5′-CCAAATTCGTTGTCATACCAGGA-3′
GAPDH probe: 5′-ATGCCAGCCCCAGCGTCAAAGGT-3′

**Surface flow cytometry analysis of PD-L1**. Cancer cells were incubated for 48 h after exposure to 10 Gy X-rays or PAPRi with or without inhibitors and then harvested for flow cytometry analysis. Adherent cells were harvested by shake-off in 1 mM EDTA-PBS without trypsinisation. Harvested cells were washed with ice-cold 1 mM EDTA-PBS and then stained with anti-PD-L1 antibodies for 20 min on ice. Dead cells detected by PI (Sigma-Aldrich) were excluded in the analysis. Flow cytometry analysis was performed on an Attune NxT Flow Cytometer (Thermo Fisher Scientific). The MFI (PD-L1–isotype) is calculated as: the MFI (PD-L1) is subtracted by the MFI (isotype control).

**Analysis of the databases**. The normalised RNA sequence and mutation status data provided by The Cancer Genome Atlas (TCGA) project were downloaded from the Genomic Data Commons Data Portal. The neoantigen data were obtained from The Cancer Immunome Atlas (http://tcia.at/). The studies for analysis were chosen based on the following two conditions: the number of cases with wild-type PD-L1 (total *N*) was more than 300 and the number of cases with mutations in each set of genes was more than 1% of the total *N*. Samples with or without mutations, including indel and point mutations, in BRCA2, PALB2, Ku70/80 or any of these four genes were compared using the Mann–Whitney *U*-test. Box plots were created using the BoxPlotR tool[60]. The samples that had the RPKM value = 0 were omitted from the log plots. The results of analyses of other tumours are shown in Supplementary Table 1.

**Statistical analysis**. Quantification of PD-L1 expression in the immuno-fluorescence study was carried out blindly with >200 cells per sample being scored. Dot density plots were created by SigmaPlot 12.0. Statistical significance was determined using Student's two-tailed *t*-test by SigmaPlot 12.0. *$P < 0.05$, **$P < 0.01$, ***$P < 0.001$.

**Data availability**. All relevant data are available from the corresponding author(s) upon reasonable request.

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

## Acknowledgements

We thank Yoshimi Omi, Akiko Shibata, Yuka Hirota, Yuka Kimura, Yoko Hayashi, Shiho Nakanishi and Mika Sato for assisting with the lab work and all the supports. We also thank S. Tomogane, S. Sato, Y. Komatsu, M. Saito, A. Sakurai and Y. Yoshimatsu for their support regarding the initial findings of IR or chemotherapeutic-agent-dependent PD-L1 upregulation. Dr. Hiro Sato appreciates Drs. T. Ohno, J.I. Saitoh, H. Kawamura, N. Kubo, T. Mizukami, A. Adachi, M. Iwanaga, S. Shiba, M. Onishi and Y. Mori for their support in the hospital work during the revision work. This work was supported by JSPS KAKENHI Grant Numbers JP26701005 and JP17H04713 to A.S., JP16H05388 to T.N., JP17K16420 and JP15K19771 to H.S., the Cell Science Research Foundation, the Daiichi Sankyo Foundation of Life Science and the Takeda Science Foundation. This work was supported by the Program of the network-type Joint Usage/Research Center for Radiation Disaster Medical Science of Hiroshima University, Nagasaki University and Fukushima Medical University. This work was also supported by Grants-in-Aid from the Ministry of Education, Culture, Sports, Science and Technology of Japan for programs for Leading Graduate Schools, Cultivating Global Leaders in Heavy Ion Therapeutics and Engineering.

## Author contributions

A.S. designed the experiments and wrote the paper. The experiments including immunoblotting, qPCR and flow cytometry were performed by H.S., A.N., T.B.M.P., Y.H., M.I., R.S., S.K. and A.S. The clonogenic survival experiment was performed by E.N. Immunofluorescence analysis were performed by Y.H., M.I., R.S. and A.S. siRNA library experiment was performed by A.S. The TCGA analysis was performed by T.Y. Acquired data were analysed and interpreted by H.S., T.Y. and A.S. The manuscript was reviewed by T.O., Y.Y., K.D.H., Y.S., K.K., K.M. and T.N. Administrative, technical or material support was provided by T.O. and T.N. The study was supervised by A.S.

## Additional information

**Competing interests:** The authors declare no competing financial interests.

