## [Peer Review File · Nature Communications]

Reviewer #1 (Remarks to the Author)

This is a clearly written piece of work that provides novel evidence that ATM, ATR and CHK1 signalling are all important in the induction of PD-L1 expression following exposure to a variety of DNA damaging agents. The studies have been carefully conducted and the data look credible. They have attempted to support the clinical relevance of their data by looking at PD-L1 expression and specified gene defects in a variety of tumours on the TCGA database. Broadly these data are supportive but are more variable and less convincing than is claimed in the text. They should acknowledge this variability and it is not entirely unexpected as PD-L1 expression is likely to be multifactorial. Nevertheless, the data do suggest that combining a DNA damaging agent with anti PD-1 therapy might be useful clinically, although not necessarily just to patients stratified according to their tumour's DDR status. Ku knockdown seems to cause the greatest increase in PD-L1 mRNA expression, although this is not accompanied by a much greater increase in the protein levels, both interesting observations that are not discussed.

There are some deficits in the work though. The authors claim that it is DNA double-strand breaks that is activating the pathway to PD-L1 expression however all the agents used will cause replicative stress and even X-rays cause around 25 single strand breaks for every DSB. Replication stress causes regions of single-stranded DNA as does repair of cisplatin adducts by nucleotide excision repair. The predicted result of Ku knockdown is also increased resection and creation of single-stranded DNA, is this the trigger rather than DSB? I feel more could be done to clarify this issue, and to support the model in Figure 6, by use of agents that almost exclusively cause DSBs, e.g. etoposide. Perhaps RPA foci could also be measured. It would also have been useful to have some understanding of the extent of DNA damage and cytotoxicity induced by the different agents at the concentrations used. A non-DNA damaging cytotoxic agent, e.g. paclitaxel, might have been useful as a control. If the authors want to increase the clinical relevance of their work it would be useful to look at the induction of PD-L1 by oxidative stress as oxygen radicals are a) higher in tumours, b) part of the inflammatory response and c) damage DNA.

I am not sure why the authors selected UCN01 as it is not specific for CHK1 see <https://www.ncbi.nlm.nih.gov/pubmed/17850214> so it is difficult to be certain that the effects are mediated by CHK1 inhibition. There are many more potent and specific inhibitors currently available, e.g. PF477736, CCT244747, MK8776, LY2603618 that could have been used.

Reviewer #2 (Remarks to the Author)

In this study, Sato and colleagues have shown that PD-L1 upregulation after IR is due in part to activation of the DSB repair system and that the activation of Chk1 following DNA end resection is a critical step leading to the upregulation of PD-L1.

This is a very interesting study from a mechanistic perspective. Unfortunately, however, there appears to be dissociation between the main mechanism behind the efficacy of PD-1 axis blockade (adoptive immune resistance) and a mechanism behind the efficacy of combined IR and PD-1 axis blockade proposed by the authors (native immune resistance).

The current - and fairly widely accepted - understanding of PD-L1 regulation in cancer is that anti-tumor T cells (and possibly other cells) produce IFN γ (and possibly other cytokines) that induce PD-L1 expression on tumor cells. Thus, tumor cells with abundant neo-antigens (such as those with defective DNA damage repair mechanisms) are more frequently recognized by anti-tumor T cells and thus exposed to more IFN γ , resulting in higher PD-L1 expression (PMID: 25583798). Along the similar line, increased mutational burden is the reason that tumors with defects in DNA damage repair are more responsive to immunotherapy.

It seems the authors believe that achieving PD-L1 expression is the goal of immunogenic radiation as well as that impairment of DNA damage repair is immunogenic (e.g. lines 99-101). Although PD-L1 expression may be a marker of anti-tumor immune response, its expression in isolation is

not the mechanism behind response to checkpoint blockade. In another word, simply enforcing PD-L1 over-expression on a tumor cell may not be sufficient to make a tumor susceptible to PD-1 axis blockade. Thus, it would be important to show that PD-L1 expressing cells secondary to IR respond to anti PD-1/PD-L1 agents in their cell line experiments. Specific comments are as follows:

1. Figure 5 is largely unconvincing in relation to the manuscript's thesis because PD-L1 expression in these tumors is likely being driven by increased anti-tumor T cell activity resulting from these tumors having more net-antigens. While the authors' cell culture work exist in isolation from immune cells, the TCGA data do not.

2. Is the high dose radiation killing the cell lines? The authors state in the results that "An irradiating dose of 10Gy is toxic and severely damages the cells; however, cell death was not observed by detecting PARP-1 cleavage in this time range" (line 205). Does this mean that the cells did die later on? Are the authors measuring PD-L1 expression (at 48hours) in cells that will soon be dead (e.g. at 120 hours)? If so, this needs to be clearly stated. The manuscript currently reads in a way that suggests the irradiated cells recover. PD-L1 expression in a tumor cell that's going to die has a different significance than PD-L1 expression in a tumor cell that will recover and go on to replicate.

The authors predominantly show PARP-1 Western Blots to prove that cells are viable. Perhaps they could provide a timeline of PARP-1 cleavage and possibly other viability markers extending to 72, 96, and 120 hours? Fig1a's long exposure looks like the U2OS cells might be about to start dying at 48 hours.

3. DSB repair generally occurs pretty quickly, on the order of minutes to hours. Why does it take so long for PD-L1 to show increased transcripts and protein? Is this related to cell death?

4. Is the increase in PD-L1 expression (approximately 2 fold from baseline by RNA and protein) relevant? The range of PD-L1 expression in inflamed and non-inflamed tissue (either in tumors or infection) seems to be much greater than two-fold when one looks at other published IHC and flow cytometry data (PMID: 24714771). Furthermore, the authors argue that flow cytometry was not performed because irradiation causes morphologic changes. Why would morphologic changes that affect flow cytometry not also affect immunofluorescence staining? It would seem the same principles apply to both. Flow cytometry for PD-L1 in cultured cells is a more standard method. The authors should make a stronger case why they chose not to use it. Better yet, show the flow data. IHC might also be helpful.

Minor issues:

1. PMID 25754329 along with PMID 24382348 should be cited to describe the recent pre-clinical studies showing that the combination of radiation with check point blockade can enhance tumor control in mice.

2. Page 23, line 369: tumor growth -> tumor growth delay

3. Reference 20 does not support what the authors described in the lines 368-370.

Reviewer #3 (Remarks to the Author)

H. Sato et al dissected the mechanism by which DSB up-regulates PD-L1 expression in cancer cells. Despite the general interest in understanding PD-L1 regulation by cancer cells and stromal/immune cells, conclusions in this study were almost exclusively based on experiments in one cell line, U2OS. In addition, no biological significance or functional implication of the mechanistic assertion was generated or explored. Alternative mechanisms, especially those involving canonical pathway regulation of PD-L1 expression, were not accounted for at all. The

degree of differential expression was marginal without statistical treatment on significance, e.g., Fig 2b-c. .

Major points:

"Notably, such upregulation 210 after IR was not observed in normal primary human fibroblast cells (Extended Data Fig. 1c)". Why wouldn't IR cause DSB in primary fibroblast cell as well?

PD-L1 mRNA upregulation in Fig 1e, 1f, 2c and 2f is marginal (~ 1.5-3 fold) considering this was qRT-PCR data. Also there was no statistics on the differential expression.

Instead of just testing if HLA is induced, the authors should test activation status of STAT1 and levels of IRF1 as upstream regulators of PD-L1 induction by the IFN-JAK-STAT pathway.

"In this study, we used immunofluorescence (IF) analysis, since IR or drug treatment causes morphological changes, for example, increases or decreases in the overall surface area of cells (see IF data in figures), which may affect total intensity that can be detected by other analyses such as flow cytometry." One should be able to set a correct cutoff of MFI for FACS based analysis (which would be more quantitative) with proper negative controls. Using IF, despite the non-permeabilization, one could not exclude the possibility of intracellular PD-L1 staining.

The WB in Figure 2A, 2B are quite weak. Overall, the PD-L1 protein and mRNA upregulation is not strong. However, will this poise the cells to be able to upregulate PD-L1 much higher in presence of type I/II interferon? This is considering that the cells with DNA damage signaling may have more immune cells and interferons in its microenvironment

The finding using siBRCA2 needs to be replicated in 1-2 more lines (at least mRNA and MFI based on IR +/- PARPi/ATMi/CHKi treatments). This is important since is already a report on enrichment of BRCA2 mutations in responders to anti-PD-1, and an increased PD-L1 would be in line with such a phenotype.

Could combined loss of Ku70/80 AND BRCA2 induce much higher PD-L1. This is relevant since there were reports showing that Ku70/80 were generally expressed lower in melanoma (Korabiowska M, Mod Pathol 2002 PMID: 11950917), and the report on BRCA2 mutations and anti PD-1 response was in melanoma.

To test if BRCA2/PALB2/Ku70/Ku80 were related to PD-L1 increase - the authors should also examine the CCLE data (based on cBioPortal, ~10% of the 1019 cell lines in Cancer Cell Line Encyclopedia (Novartis/Broad, Nature 2012) have BRCA2 mutations and there are a few more with PALB2 or Ku70/80. Alternatively, the authors can also test the expression levels). This way, the authors will be able to distinguish high cancer cell intrinsic PD-L1 correlation with the mutation/expression loss of these genes. In the bulk tumor, there will be multiple variable which may drive PD-L1 upregulation (e.g IFN) which need not be related to the DSB response.

Minors:

Page 21, line 351, "In contrast"

Page 23, line 368, ".although it is evident that the blockage of PD-1/PD-L1 interaction enhances in vivo tumour growth in combination with IR19,20." Should "growth" be "suppression"?

"In previous studies, PD-1 therapy was shown to achieve remarkable clinical responses in patients with several types of cancer; however, approximately 30% of patients still remained poorly responsive2-4." The percentage of anti PD-1 responsive patients is more likely to be in 20-40% range based on these references (melanoma, RCC and NSCLC)

Reviewer #1 (Remarks to the Author):

This is a clearly written piece of work that provides novel evidence that ATM, ATR and CHK1 signalling are all important in the induction of PD-L1 expression following exposure to a variety of DNA damaging agents. The studies have been carefully conducted and the data look credible. They have attempted to support the clinical relevance of their data by looking at PD-L1 expression and specified gene defects in a variety of tumours on the TCGA database. Broadly these data are supportive but are more variable and less convincing than is claimed in the text. They should acknowledge this variability and it is not entirely unexpected as PD-L1 expression is likely to be multifactorial.

We appreciate the time and effort that reviewer #1 has expended in understanding the significance of this study. Following the suggestion, the Introduction section was largely revised by adding references to several primary studies and multiple roles of PD-1/PD-L1 in cellular responses.

Nevertheless, the data do suggest that combining a DNA damaging agent with anti PD-1 therapy might be useful clinically, although not necessarily just to patients stratified according to their tumour's DDR status.

Thank you for this comment, with which we completely agree. We corrected our statement as suggested (page 32, lines 6-7).

Ku knockdown seems to cause the greatest increase in PD-L1 mRNA expression, although this is not accompanied by a much greater increase in the protein levels, both interesting observations that are not discussed.

Indeed, not only in the case of Ku80 depletion, we found that there is not a perfect correlation between PD-L1 mRNA and protein levels after DNA damage. Recently, Li CW et al. revealed that PD-L1 is regulated by proteasome-dependent degradation (Li CW et al., Nat. Comm., 2017; PMID: 27572267). They showed that GSK3 β induces the phosphorylation-dependent proteasome degradation of PD-L1 via the β -TrCP complex. In fact, our immunoblotting analysis always showed a smeared band. For example, the smeared bands were evident after Etp treatment (Figure 5e). This suggests that further post-translational modifications may be induced for the fine-tuning of PD-L1 protein

levels after DNA damage. Therefore, we speculate that PD-L1 protein may be regulated not only by the activation of transcription but also by multiple post-translation modifications, e.g. ubiquitination, after DNA damage. Text on this issue has been added to the manuscript (page 30, lines 8-15).

There are some deficits in the work though. The authors claim that it is DNA double-strand breaks that is activating the pathway to PD-L1 expression however all the agents used will cause replicative stress and even X-rays cause around 25 single strand breaks for every DSB. Replication stress causes regions of single-stranded DNA as does repair of cisplatin adducts by nucleotide excision repair. The predicted result of Ku knockdown is also increased resection and creation of single-stranded DNA, is this the trigger rather than DSB? I feel more could be done to clarify this issue, and to support the model in Figure 6, by use of agents that almost exclusively cause DSBs, e.g. etoposide. Perhaps RPA foci could also be measured.

Thank you for this comment. To confirm the findings that PD-L1 is upregulated in response to DSBs, the expression of PD-L1 was examined in cells treated with etoposide (Etp). Similar to the results after X-rays, the treatment with Etp upregulated PD-L1 expression (Fig. 1e, 1f and 1h). These data strongly support the notion that DSBs upregulate PD-L1 expression. In addition, we showed that Ku80 depletion significantly enhanced Etp-dependent PD-L1 upregulation, which was suppressed by treatment with a Chk1 inhibitor, MK8776 (Fig. 5e, 5i and 5j). The increase in resection following Ku80 depletion was identified by counting the number of RPA foci after Etp treatment (Fig. 5f).

>Replication associated DSB

Recent studies including our own work demonstrate that Ku can bind to DSB ends not only at two-ended DSBs, but also at one-ended DSBs (also called single ended DSB), prior to HR (Biehs et al. and Shibata*, Mol. Cell, 2017; PMID: 28132842, Chanut et al., Nat. Comm., 2016; PMID: 28607502). As supplemental information for the reviewer, I have attached my review paper in this letter (Shibata, Mut. Res., in press). Thus, although we use the term 'replicative stress' in this manuscript, chemotherapeutic agents can (or highly likely) allow Ku binding at DSB ends because they induce one-ended DSBs during DNA replication. As discussed in previous literature including our work, we hypothesise that the Ku binding is important to protect DNA ends until the

repair pathway is directed towards HR. Thus, we think that Ku-dependent DSB end protection is important to suppress the excessive Chk1 activation and PD-L1 expression not only at direct DSB ends but also at indirect DSB ends via the replication fork. However, it may be too complicated for general readers if we bring the notion; therefore, we did not state the concept in terms of Ku binding at replication-associated DSBs in the model figure. Nonetheless, if the reviewer feels that we should add the diagram of one-ended DSBs as well as two-ended DSBs, the model figure will be modified as suggested.

It would also have been useful to have some understanding of the extent of DNA damage and cytotoxicity induced by the different agents at the concentrations used.

The results of γ H2AX foci with the time course and the cell survival assay (colony assay) after IR, Etp, CPT or APH in U2OS cells are shown in Extended Data Fig. 2a-c.

A non-DNA damaging cytotoxic agent, e.g. paclitaxel, might have been useful as a control.

We appreciate this suggestion, and we understand that an examination of a non-DNA-damaging agent would be informative. Therefore, we tested whether the treatment of paclitaxel affects PD-L1 in U2OS, H1299 and DU145 cells. As expected, we did not see any upregulation of PD-L1 in U2OS cells (Figure L1 in this letter). Rather, the paclitaxel reduced the PD-L1 expression in H1299 and DU145 cells (Figure L1 in this letter). Although our analysis indicated no upregulation of PD-L1 by the paclitaxel treatment, we are reluctant to include these data in this manuscript because another group has published a paper showing that paclitaxel upregulates PD-L1 in other cancer cell lines (Peng J., Can. Res., 2015, PMID: 26573793). Although we do not have any idea to explain about this discrepancy, we do not think this affects our conclusion. Therefore, we omitted the paclitaxel data in this manuscript. I hope for your understanding on this issue.

If the authors want to increase the clinical relevance of their work it would be useful to look at the induction of PD-L1 by oxidative stress as oxygen radicals are a) higher in tumours, b) part of the inflammatory response and c) damage DNA.

Thank you for this comment. We completely understand the importance of investigating the PD-L1 response after oxygen radical treatment in our assay. Because an important finding in the context of ROS production and PD-1 therapy was recently published (Chamoto K et al., PNAS, 2017; PMID: 28096382), we speculate that ROS in the tumour

environment may also be a key factor regulating PD-L1 expression via DNA damage.

As the referee would expect, oxygen radicals induce SSB and base damage, but they do not directly induce DSBs unless a high dose is used. For example, we have previously shown that 500 μ M H₂O₂ is required to induce ~5 DSBs per cell (Lobrich, Shibata et al., Cell Cycle, 2010; PMID: 20139725). Nevertheless, PD-L1 may be upregulated by DNA damage response after its exposure to oxygen radicals, particularly when SSB and/or base damage repair is defective. Although the genes involved in SSB/BER were not included in the siRNA screen of this study, we think that the idea is very interesting and important. Hence, we would like to address this in future work. Thank you for your understanding.

I am not sure why the authors selected UCN01 as it is not specific for CHK1 see <https://www.ncbi.nlm.nih.gov/pubmed/17850214> so it is difficult to be certain that the effects are mediated by CHK1 inhibition. There are many more potent and specific inhibitors currently available, e.g. PF477736, CCT244747, MK8776, LY2603618 that could have been used.

To confirm the results demonstrating that PD-L1 upregulation requires Chk1 activity, we used MK8776. We found that MK8776 inhibits PD-L1 upregulation after DNA damage (Fig. 2e (immunoblotting), 2f (qPCR), 2g (flow cytometry), 4h-k (siBRCA2) and 5j (siKu80)).

Reviewer #2 (Remarks to the Author):

In this study, Sato and colleagues have shown that PD-L1 upregulation after IR is due in part to activation of the DSB repair system and that the activation of Chk1 following DNA end resection is a critical step leading to the upregulation of PD-L1.

This is a very interesting study from a mechanistic perspective. Unfortunately, however, there appears to be dissociation between the main mechanism behind the efficacy of PD-1 axis blockade (adoptive immune resistance) and a mechanism behind the efficacy of combined IR and PD-1 axis blockade proposed by the authors (native immune resistance).

The current - and fairly widely accepted - understanding of PD-L1 regulation in cancer is that anti-tumor T cells (and possibly other cells) produce IFN γ (and possibly other cytokines) that induce PD-L1 expression on tumor cells. Thus, tumor cells with abundant neo-antigens (such as those with defective DNA damage repair mechanisms) are more frequently recognized by anti-tumor T cells and thus exposed to more IFN γ , resulting in higher PD-L1 expression (PMID: 25583798). Along the similar line, increased mutational burden is the reason that tumors with defects in DNA damage repair are more responsive to immunotherapy.

We apologise that we did not sufficiently explain the background of the research field and did not adequately acknowledge previous studies. In the revised manuscript, we have largely revised the Introduction section and have cited additional important literature in this section. As suggested by reviewer #2, we completely agree with the notion that the upregulation of neo-antigens due to a high level of mutational load triggers T-cell activation and IFN γ release, which results in high PD-L1 expression, in the tumour microenvironment (from page 5, line 17 to page 6, line 13 and page 25, lines 5-8).

In contrast, our study demonstrates the molecular mechanism of PD-L1 upregulation in response to DNA damage and that PD-L1 upregulation requires DNA damage signalling in living cells. We have clarified this point throughout the revised manuscript, i.e. the difference between PD-L1 upregulation due to high mutational load, which is a consequence of DNA repair defect resulting in mutations, and PD-L1 upregulation caused by DNA damage signals, which occur during DNA repair (please see below).

It seems the authors believe that achieving PD-L1 expression is the goal of immunogenic radiation as well as that impairment of DNA damage repair is immunogenic (e.g. lines 99-101). Although PD-L1 expression may be a marker of anti-tumor immune response, its expression in isolation is not the mechanism behind response to checkpoint blockade. In another word, simply enforcing PD-L1 over-expression on a tumor cell may not be sufficient to make a tumor susceptible to PD-1 axis blockade. Thus, it would be important to show that PD-L1 expressing cells secondary to IR respond to anti PD-1/PD-L1 agents in their cell line experiments.

Thank you for your suggestion. We understand that the text in the original manuscript may be misleading. To clarify the purpose of this study, i.e. the elucidation of the molecular mechanism of PD-L1 expression in response to DNA damage, we have modified the text as follows.

- 1) We discovered that PD-L1 is upregulated by DNA damage signalling, i.e. ATM/ATR/Chk1.
- 2) This upregulation is induced through the IRF1 pathway. (this is shown in our new data suggested by reviewer #3).
- 3) In the Discussion section, we avoided making a statement describing that our finding contributes to the improvement of PD-1 therapy. Rather, we have stated the following (page 32, lines 6-12): *we propose that understanding the signalling pathway underlying the response to DSB induction by radio/chemotherapy, which leads to PD-L1 upregulation, is of high importance to develop prognostic molecular markers and PD-1 blockade cancer immunotherapy. The factors could be pharmaceutical targets to influence the therapeutic efficacy when combined PD-1 and radio/chemotherapy is administered.*

In addition, we have deleted the sentences of *lines 99-101* as suggested.

Specific comments are as follows:

1. Figure 5 is largely unconvincing in relation to the manuscript's thesis because PD-L1 expression in these tumors is likely being driven by increased anti-tumor T cell activity resulting from these tumors having more net-antigens. While the authors' cell culture work exist in isolation from immune cells, the TCGA data do not.

We appreciate this suggestion. Following the reconsideration of the novelty of this study and the interpretation of the TCGA analysis, we now consider that PD-L1 upregulation

in tumours relies more on the increase in neo-antigens in cancer cells harbouring mutations in DNA repair genes. However, we still speculate that the upregulation of PD-L1 might be partly caused by DNA damage signalling in repair-deficient cancer cells.

In this revision, we analysed the levels of neo-antigens in the tumours harbouring mutations in the DNA repair genes. As expected, a high level of neo-antigens per sample was observed in the DNA repair-defective samples from breast and colorectum; however, samples from stomach and uterus did not show a statistical increase in the levels of neo-antigens by the mutation in the DNA repair genes (Extended Data Fig. 11). This may support the idea that additional factors (including DNA damage in DNA repair-defective cancer cells) may be involved in PD-L1 upregulation.

Therefore, in the revised manuscript, we have added the following text (page 31, lines 12-18): *This dataset analysis may not represent a direct correlation between PD-L1 expression and DSB signalling because the enhancement might largely be due to the high expression of neoantigens. However, because the statistical significance of the enhancement of neoantigens is less than that of the enhancement of PD-L1, we speculate that DNA damage signalling may partly contribute to the increase in PD-L1 expression in tumours when DNA repair and signalling are downregulated, particularly in patients who are treated with radio/chemotherapy.* We retained the data on TCGA analysis but moved it to Extended Data Figs. 9-11 in the revised manuscript. Thank you for your comments and understanding.

2. Is the high dose radiation killing the cell lines? The authors state in the results that “An irradiating dose of 10Gy is toxic and severely damages the cells; however, cell death was not observed by detecting PARP-1 cleavage in this time range” (line 205). Does this mean that the cells did die later on?

We have now added the data of the colony formation assay, which is the most standard assay to measure cell viability after DNA damage (Extended Data Fig. 2c). The data suggest that >90% cells died or at least exhibited arrested cell growth, for example, cellular senescence, at ~14 days after exposure to 10 Gy X-rays.

Are the authors measuring PD-L1 expression (at 48hours) in cells that will soon be dead (e.g. at 120 hours)?

If so, this needs to be clearly stated. The manuscript currently reads in a way that suggests the irradiated cells recover. PD-L1 expression in a tumor cell that’s going to die has a different significance than PD-L1 expression in a tumor cell that will recover and

go on to replicate.

Thank you for this suggestion, but indeed, it is very difficult to answer the timing of cell death after IR because irradiated cells do not synchronously undergo cell death.

The consensus in radiation biology is that irradiated cells die following one to two cell cycle divisions or stop growing because of cellular senescence. The proportion of cell death and senescence is variable and is dependent on the condition, e.g. IR doses or cell type. But, as shown in Extended Data Fig. 2c, it is clear that cells that were irradiated with >90% of 10 Gy die or stop growing within ~14 days after IR. In the revised manuscript, we have added this information. In addition, to answer the question whether cells that recovered from IR maintain PD-L1 upregulation, we examined the level of PD-L1 expression in cells that could regrow after an exposure to a dose of 10 Gy (Extended Data Fig. 2d). Consistent with the observation in a previous report (Wu CT et al., Scientific Reports, 2016; PMID: 26804478), the level of PD-L1 returned back to normal in the recovered cells

The authors predominantly show PARP-1 Western Blots to prove that cells are viable.

In our immunoblot analysis, because we collected only adherent cells following washing with PBS before harvesting cells, we did not see obvious PARP-1 cleavage. We also examined the level of caspase3 cleavage in our samples for immunoblotting analysis; however, no clear caspase3 cleavage was observed (Figure L2 in this letter).

Although the reviewer might have a concern about that we may have detected a PD-L1 signal derived from dead or dying cells, notably, the number of dead cells did not always correlate with PD-L1 upregulation in our immunoblotting analysis. For example, in Figure 2d (revised manuscript), PD-L1 expression was significantly suppressed by the inhibition of ATR/Chk1 activity, although cells treated with ATR or Chk1 inhibitor exhibited greater PARP-1 cleavage. Thus, the present data suggest that PD-L1 upregulation is regulated by DNA damage signalling in living cells rather than from dead cells.

In addition, as suggested by this referee and reviewer #3, we performed flow cytometry analysis. In this analysis, PI-positive dead cells were excluded. Consistent with the data obtained by using immunoblotting analysis, we found that PD-L1 is upregulated in a Chk1-dependent manner after DNA damage. We also detected PD-L1 mRNA upregulation using qPCR. mRNA expression is unlikely to be induced by dead cells. Thus, we believe that PD-L1 upregulation after DNA damage is dependent on the

signalling from living cells. This issue is clarified in the revised manuscript (page 18, lines 9-11; page 25, line 16-page 26, line 2; and Extended Data Fig. 2 legend) and thank you for your comment.

3. DSB repair generally occurs pretty quickly, on the order of minutes to hours. Why does it take so long for PD-L1 to show increased transcripts and protein? Is this related to cell death?

For the revised version of this manuscript, to determine the timing of PD-L1 mRNA upregulation, we carried out a short time course experiment, namely, at 2, 4, 8, 16 and 24 h after IR (Extended Data Fig. 3e). Notably, PD-L1 mRNA increased from 2 h after IR, at which point there was ongoing DSB repair (Extended Data Fig. 2a). As requested by reviewer #1, we examined the level of DSBs by monitoring γ H2AX foci, a marker of DSB. As shown in Extended Data Fig. 2a-b, γ H2AX foci still remained at 2–48 h after exposure to 10 Gy.

In terms of repair kinetics after IR, most DSBs in irradiated G1-cells are quickly repaired by NHEJ (Shibata et al., EMBO J, 2011; PMID: 21317870). In contrast, DSBs in irradiated S/G2 cells undergo HR, which is a slow repair pathway compared with NHEJ (Shibata et al., EMBO J, 2011; PMID: 21317870). In this study, we show that ATR/Chk1 activation is important to transduce the damage signal towards PD-L1 upregulation. Because ATR/Chk1 is activated at ssDNA regions during HR at the slow phase of DSB repair (Rhind N, Molecular Cell, 2009 ; PMID: 19328060), we think that the timings of PD-L1 expression (>16–24 h) and repair kinetics (<24–48 h) are not entirely uncorrelated.

In addition, as a question raised by reviewer #1 regarding the imperfect correlation between mRNA and PD-L1 protein, PD-L1 proteins might be regulated by post-transcriptional regulations (Li CW et al., Nat. Comm., 2017; PMID: 27572267). Text describing this possibility was added in the Discussion section (page 30, lines 8-15).

4. Is the increase in PD-L1 expression (approximately 2 fold from baseline by RNA and protein) relevant? The range of PD-L1 expression in inflamed and non-inflamed tissue (either in tumors or infection) seems to be much greater than two-fold when one looks at other published IHC and flow cytometry data (PMID: 24714771).

In the same set of experiments by both immunoblotting and flow cytometry, we compared the magnitude of PD-L1 upregulation between IR and IFN γ treatments. Our analysis

showed that 10 Gy (48 h) is comparable to treatment with 1–5 ng/mL IFN γ treatment (24 h) (Extended Data Figure 4a-b).

Furthermore, the authors argue that flow cytometry was not performed because irradiation causes morphologic changes. Why would morphologic changes that affect flow cytometry not also affect immunofluorescence staining? It would seem the same principles apply to both. Flow cytometry for PD-L1 in cultured cells is a more standard method. The authors should make a stronger case why they chose not to use it. Better yet, show the flow data. IHC might also be helpful.

We performed flow cytometry analysis to confirm our major findings about a) PD-L1 upregulation after DNA damage (Figure 1h), Chk1 dependence (Figure 2g), BRCA2 and Ku80 siRNA cells (Figures 4, 5) and IRF1 dependence (Figure 6).

Minor issues:

1. PMID 25754329 along with PMID 24382348 should be cited to describe the recent pre-clinical studies showing that the combination of radiation with check point blockade can enhance tumor control in mice.

These papers were cited in the introduction. We appreciate your suggestions.

2. Page 23, line 369: tumor growth -> tumor growth delay

The sentence was corrected as suggested.

3. Reference 20 does not support what the authors described in the lines 368-370.

We replaced the citation to (Dovedi, S. J. et al., Cancer research, 2014; PMID: 25274032). Thank you so much for your advice.

Reviewer #3 (Remarks to the Author):

H. Sato et al dissected the mechanism by which DSB up-regulates PD-L1 expression in cancer cells. Despite the general interest in understanding PD-L1 regulation by cancer cells and stromal/immune cells, conclusions in this study were almost exclusively based on experiments in one cell line, U2OS.

To consolidate our findings, we examined PD-L1 expression in H1299 and DU145 cells with or without DNA damage. Notably, we obtained similar results on the following: a) Chk1 dependence, b) Ku80 and c) BRCA2 in H1299 cells. However, we prefer to keep the results of U2OS cells throughout the manuscript because, as described in the original manuscript, the U2OS cell line is well utilised in the field of DSB repair because this cell line has mostly intact DNA repair and signalling, which is not the case with the other cancer cell lines.

In addition, no biological significance or functional implication of the mechanistic assertion was generated or explored.

Alternative mechanisms, especially those involving canonical pathway regulation of PD-L1 expression, were not accounted for at all.

Thank you for this comment. Following this suggestion, we investigated how the DNA-damage-dependent upregulation involves the canonical pathway by referring to latest important findings (Garcia-Diaz A et al., Cell Rep., 2017; PMID: 28494868, Shin DS et al., Can. Discov., 2017; PMID: 27903500). The details of these experiments are described below.

In addition, following the suggestions from reviewer #2, we clarified the novelty of this study in the revised manuscript, i.e. we discovered that DSBs upregulate PD-L1 in cancer cells and demonstrated its molecular mechanism. We believe that this is an important finding that significantly contributes to the understanding of the PD-L1 regulation in response to DNA damage after radio/chemotherapy.

The degree of differential expression was marginal without statistical treatment on significance, e.g., Fig 2b-c.

We apologise that statistical analysis was not performed for all of the experiments. In the revised manuscript, asterisks reflecting the statistical results were added for all of

the mRNA and flow cytometry experiments.

Major points:

"Notably, such upregulation 210 after IR was not observed in normal primary human fibroblast cells (Extended Data Fig. 1c)". Why wouldn't IR cause DSB in primary fibroblast cell as well?

To further confirm the responsiveness of PD-L1 expression after DNA damage in human primary fibroblast cells, we examined its expression after exposure to high-dose X-ray, CPT and Etp. However, similar to the data obtained in the original manuscript, we did not find clear PD-L1 upregulation after IR, CPT and Etp (Figure L3 in this letter). Further, because the PD-L1 gene might be silenced due to epigenetic modification or chromatin compaction at the promoter region in primary skin fibroblast cells, we added an HDAC inhibitor to see whether the expression was relieved or not. We found that PD-L1 expression was slightly restored compared with that without treatment; however, this was still a subtle effect compared with the upregulation by IFN γ treatment (Figure L3). We have recently published a paper showing the effect of HDAC inhibitor on NKG2D ligand expression after IR (Nakajima et al., *Oncol. Rep.*, 2017; PMID: 28677817). In this study, we found that NKG2DL was not efficiently induced in primary fibroblast cells after IR even in the presence of HDAC inhibitor. Although the mechanism has not yet been revealed, we speculate that there might be a factor in skin fibroblast cells that suppressed the signal for the expression of the ligands, which may be restored by IFN γ . However, we would like to not draw a strong conclusion on this issue using the available data; therefore, in the revised manuscript, we have stated that 'we did not observe PD-L1 upregulation in primary fibroblast cells (48BR) after DNA damage in the time range of our analysis (Extended Data Fig. 3a)' (page 14, line 17 – page 15, line 1).

PD-L1 mRNA upregulation in Fig 1e, 1f, 2c and 2f is marginal (~ 1.5-3 fold) considering this was qRT-PCR data. Also there was no statistics on the differential expression.

We performed the statistical analysis for all the qPCR and flow cytometry analyses. Thank you for your suggestion.

In addition, we have added the text regarding the possibility that PD-L1 protein levels are also regulated by the ubiquitin-dependent proteasome pathway (Li CW et al., *Nat. Comm.*, 2017; PMID: 27572267) (also please see the comment to reviewer #1). This

may be involved in the lesser increase expression of PD-L1 mRNA compared with that of PD-L1 protein.

Instead of just testing if HLA is induced, the authors should test activation status of STAT1 and levels of IRF1 as upstream regulators of PD-L1 induction by the IFN-JAK-STAT pathway.

We appreciate this suggestion. We examined the phosphorylation status of STAT1, STAT3 and IRF1 after IR, CPT or Etp in U2OS and H1299 cells (Fig. 6a and 6b). Notably, we found an increase in STAT1/3 phosphorylation and IRF1 upregulation after IR in these cells. Further, we showed that the depletion of IRF1 in these cells reduces PD-L1 upregulation after IR (Fig. 6c, f and g).

"In this study, we used immunofluorescence (IF) analysis, since IR or drug treatment causes morphological changes, for example, increases or decreases in the overall surface area of cells (see IF data in figures), which may affect total intensity that can be detected by other analyses such as flow cytometry." One should be able to set a correct cutoff of MFI for FACS based analysis (which would be more quantitative) with proper negative controls. Using IF, despite the non-permeabilization, one could not exclude the possibility of intracellular PD-L1 staining.

Reviewer #2 also requested that flow cytometry analysis be performed. We carried out this flow cytometry analysis for experiments, which strengthened our conclusions (Fig. 1h, 2g, 4f-k, 5h-j, 6f-g).

The WB in Figure 2A, 2B are quite weak. Overall, the PD-L1 protein and mRNA upregulation is not strong. However, will this poise the cells to be able to upregulate PD-L1 much higher in presence of type I/II interferon? This is considering that the cells with DNA damage signaling may have more immune cells and interferons in its microenvironment

Thank you for raising this interesting idea. We examined the magnitude of PD-L1 upregulation by immunoblotting and flow cytometry in the presence of IFN γ (Extended Data Fig. 4a-b). Although some additional increase was observed in the presences of IFN γ , it did not seem to be a synergistic increase. This idea will be further investigated in an *in vivo* model in the future.

The finding using siBRCA2 needs to be replicated in 1-2 more lines (at least mRNA and MFI based on IR +/- PARPi/ATMi/CHKi treatments). This is important since is already a report on enrichment of BRCA2 mutations in responders to anti-PD-1, and an increased PD-L1 would be in line with such a phenotype.

Following the suggestions, qPCR and flow cytometry analyses were carried out in H1299 cell lines +/- BRCA2 siRNA (Fig. 4g, 4j-k, and Extended Data Fig. 6c). Notably, consistent with the results in U2OS cells, we found that the depletion of BRCA2 enhanced PD-L1 upregulation after IR or PARP inhibition in H1299 as well as U2OS cells. In addition, the results showed that the PD-L1 upregulation is Chk1-dependent (Fig. 4h-k and Extended Data Fig. 6c).

Could combined loss of Ku70/80 AND BRCA2 induce much higher PD-L1. This is relevant since there were reports showing that Ku70/80 were generally expressed lower in melanoma (Korabiowska M, Mod Pathol 2002 PMID: 11950917), and the report on BRCA2 mutations and anti PD-1 response was in melanoma.

We applied the double knockdown (data not shown), but the downregulation of both DSB repair pathways (NHEJ and HR) was lethal. We added a discussion about how low expression of Ku80 and/or BRCA2 affects PD-L1 expression in melanoma cells (page 30, line 15- page 31, line 1). Thank you for your suggestions.

To test if BRCA2/PALB2/Ku70/Ku80 were related to PD-L1 increase - the authors should also examine the CCLE data (based on cBioPortal, ~10% of the 1019 cell lines in Cancer Cell Line Encyclopedia (Novartis/Broad, Nature 2012) have BRCA2 mutations and there are a few more with PALB2 or Ku70/80. Alternatively, the authors can also test the expression levels). This way, the authors will be able to distinguish high cancer cell intrinsic PD-L1 correlation with the mutation/expression loss of these genes. In the bulk tumor, there will be multiple variable which may drive PD-L1 upregulation (e.g IFN) which need not be related to the DSB response.

Reviewer #2 suggested that the data of TCGA analysis might have been misinterpreted; that is, PD-L1 upregulation in tumour samples may rely more on the increase in neo-antigens in cancer cells harbouring mutations in DNA repair genes. Therefore, we moved the TCGA data to supplemental materials, and we avoided drawing any strong

conclusion that constitutive DNA damage in tumours may upregulate PD-L1 in the text on the bioinformatic analysis. Thank you for your understanding.

Minors:

Page 21, line 351, "In contrast"

This misspelling was corrected. Thank you.

Page 23, line 368, "..although it is evident that the blockage of PD-1/PD-L1 interaction enhances in vivo tumour growth in combination with IR19,20." Should "growth" be "suppression"?

This was also corrected.

"In previous studies, PD-1 therapy was shown to achieve remarkable clinical responses in patients with several types of cancer; however, approximately 30% of patients still remained poorly responsive²⁻⁴." The percentage of anti PD-1 responsive patients is more likely to be in 20-40% range based on these references (melanoma, RCC and NSCLC)

Thank you for your suggestion. We corrected the percentage as suggested.

Reviewers' Comments:

Reviewer #2 (Remarks to the Author)

The authors have satisfactorily responded to the reviewers' comments for the most part. I have a couple of minor suggestions, however.

1. Upregulation of PD-L1 on tumor cells by DNA damage signaling (ATM/ATR/Chk1) through the IRF1 pathway likely corresponds to type III tumor microenvironment (intrinsic induction) among the four types to tailoring cancer immunotherapeutic modules suggested by Teng and colleagues (PMID 25977340). The authors should make it more evident that what they have found explains the upregulation of PD-L1 on tumor cells after radiation and/or chemotherapy, but it is not a rationale for combined PD-L1 blockade and radio/chemotherapy.

2. The response rate of PD-1/PD-L1 blockade (as a single agent) in various tumors is up to 45% (pembrolizumab for treatment of NSCLC as the first line therapy) meaning that the majority will not respond to the therapy.

Reviewer #2 (Remarks to the Author):

1. Upregulation of PD-L1 on tumor cells by DNA damage signaling (ATM/ATR/Chk1) through the IRF1 pathway likely corresponds to type III tumor microenvironment (intrinsic induction) among the four types to tailoring cancer immunotherapeutic modules suggested by Teng and colleagues (PMID 25977340).

Thank you for your comments.

Regarding categorization, we agree with Referee#2 that the situation showing PD-L1 upregulation after DSBs can be included in type III tumour microenvironments; however, because IR also causes inflammatory response, the signature of type I may be contained when it occurs *in vivo*. In addition, our *in vitro* analysis revealed that DSB-dependent PD-L1 upregulation requires STATs-IRF1 signals without IFN γ addition. This suggests that PD-L1 can be upregulated in the absence of TIL *in vivo*, but it uses the intrinsic STATs-IRF1 pathway. Overall, we believe that regulation after exogenous stresses such as DNA damage is not simply categorized and should be more complicated *in vivo*. Because this is an important open question, we would like to more carefully consider and discuss this issue, and hopefully write a review after this manuscript is published.

The authors should make it more evident that what they have found explains the upregulation of PD-L1 on tumor cells after radiation and/or chemotherapy, but it is not a rationale for combined PD-L1 blockade and radio/chemotherapy.

In the first revision, we have already amended the statement and we did not conclude that upregulation contributes to the enhanced efficacy in the combined PD-L1 blockage and radio/chemotherapy. However, to avoid any confusion and misunderstanding among the readers, we have added a sentence on page 26, lines 12-13.

2. The response rate of PD-1/PD-L1 blockade (as a single agent) in various tumors is up to 45% (pemrolizumab for treatment of NSCLC as the first line therapy) meaning that the majority will not respond to the therapy.

We completely agree that there are tumours, which do not respond to the PD-1/PD-L1 therapy when these are used as single agents. Thank you for your comment.